# WHAT'S IN MY HUMAN FEEDBACK? LEARNING INTERPRETABLE DESCRIPTIONS OF PREFERENCE DATA

**Rajiv Movva**[*]
UC Berkeley

**Smitha Milli**
FAIR at Meta

**Sewon Min**
UC Berkeley

**Emma Pierson**
UC Berkeley

## ABSTRACT

Human feedback can alter language models in unpredictable and undesirable ways, as practitioners lack a clear understanding of what feedback data encodes. While prior work studies preferences over certain attributes (e.g., length or sycophancy), automatically extracting relevant features without pre-specifying hypotheses remains challenging. We introduce *What's In My Human Feedback?* (WIMHF), a method to explain feedback data using sparse autoencoders. WIMHF characterizes both (1) the preferences a dataset is capable of measuring and (2) the preferences that the annotators actually express. Across 7 datasets, WIMHF identifies a small number of human-interpretable features that account for the majority of the preference prediction signal achieved by black-box models. These features reveal a wide diversity in what humans prefer, and the role of dataset-level context: for example, users on Reddit prefer informality and jokes, while annotators in HH-RLHF and PRISM disprefer them. WIMHF also surfaces potentially unsafe preferences, such as that LMArena users tend to vote against refusals, often in favor of toxic content. The learned features enable effective *data curation*: re-labeling the harmful examples in Arena yields large safety gains (+37%) with no cost to general performance. They also allow fine-grained *personalization*: on the Community Alignment dataset, we learn annotator-specific weights over subjective features that improve preference prediction. WIMHF provides a human-centered analysis method for practitioners to better understand and use preference data.

**Code:** https://github.com/rmovva/wimhf ⬤

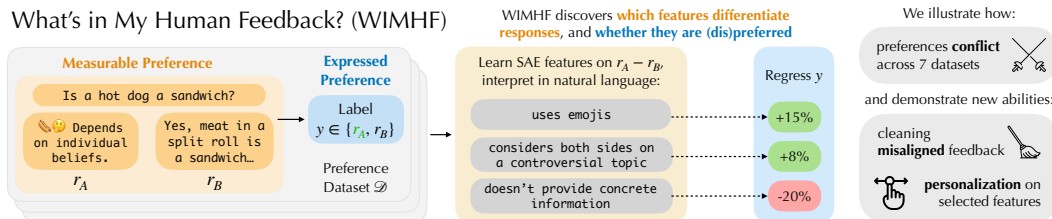

Figure 1: What's In My Human Feedback enables automated discovery of preferences from feedback data. We first discover **measurable preferences**: consistent differences within a pair of responses $(r_A, r_B)$, like "emoji usage," learned by a sparse autoencoder (SAE). Regressing the chosen response $y$ on these features yields **expressed preferences**, like "win-rate is 15% higher with emojis."

## 1 INTRODUCTION

Preference data are the foundation of language model alignment, and yet we lack a clear understanding of what they encode. Given a pair of candidate responses to a prompt, humans are asked to select the better one, and these labels are used for preference finetuning (PFT). Due to the subjectivity of this task, it is difficult to predict how human feedback will shape models: prior work shows

---

[*]Correspondence: rmovva@berkeley.edu, emmapierson@berkeley.edu.

that PFT can produce several benefits (Bai et al., 2022), but also unintentional behaviors like sycophancy or overconfidence (Sharma et al., 2023; Zhou et al., 2024). Understanding what preferences encode would enable model developers to better curate data and steer models with fewer undesirable effects.

However, describing preference data is difficult—reward models, for example, can accurately predict which of two responses a judge will prefer, but they do not yield insight as to why. Therefore, another line of work specifies simple features (politeness, humor, etc.) that are hypothesized to influence judgment, and then empirically measures whether they are preferred (Go et al., 2024; Li et al., 2025). While useful, pre-specifying features constrains what can be discovered. There are many idiosyncrasies in human feedback that may be unexpected (Tversky & Kahneman, 1974; Hosking et al., 2024), especially as pairwise ranking enters new and more specialized domains (Zhao et al., 2025; Chi et al., 2025). We therefore require an approach that enables automated discovery, from data, of important features.

Towards this goal, we propose *What's In My Human Feedback?*[1] (WIMHF), a method to explain preference datasets automatically, without pre-specifying hypotheses (Figure 1). WIMHF first learns a list of features, using a sparse autoencoder (SAE), that capture how responses in a pair consistently differ from one another. These are a dataset's **measurable preferences**: features that vary across the two responses that can, in theory, shape what the model learns. Features are fine-grained, interpretable concepts like "answers directly without clarifying questions" or "uses emojis." We use these features to study **expressed preferences**: which features actually explain the preference labels. Notably, the sparse features capture the majority of the available signal in predicting preference, achieving 84% of the signal that is predictable using dense text embeddings, and 67% of a black-box reward model. These features also match annotator-written explanations, pass qualitative validation, and outperform a prior baseline (Findeis et al., 2025).

Using WIMHF, we shed new light on the contents of seven widely-used feedback datasets, with implications for how to construct and use datasets. We show that measurable preferences depend heavily on how responses are generated: for example, the standard approach of high-temperature sampling (Bai et al., 2022; Kirk et al., 2024) produces differences in style, tone, and refusal, while a dataset that explicitly prompts for diverse responses (Zhang et al., 2025a) contains more topic-related differences (e.g., luxury vs. budget recommendations). In terms of expressed preferences, datasets often encode conflicting preferences: for example, on safety-related issues, annotators in HH-RLHF prefer deflecting unethical requests, while the strongest preference in LMArena is *against* refusals. This suggests that the common practice of mixing datasets when performing alignment (e.g., Dong et al. (2024), Ivison et al. (2024)) may encode contradictory signals.

WIMHF enables model developers to act on these findings. We use the learned features to steer model behavior via *data curation*: on LMArena, cleaning examples with the anti-refusal feature substantially improves safety (+37%) on RewardBench 2 with no cost to overall performance. The features are also levers for *personalization*: on Community Alignment, with just a few examples per annotator, we learn user-specific coefficients that improve heldout preference prediction. Importantly, unlike black-box methods, we can restrict tuning to select attributes, enabling users to, e.g., receive paragraphs instead of bullet points, while avoiding ideological echo chambers (Kirk et al., 2023). Ultimately, WIMHF provides a tractable framework for practitioners to understand human feedback, and enables new possibilities for data-centric preference learning.

## 2 EXPLAINING HUMAN FEEDBACK DATASETS

A preference dataset $\mathcal{D}$ consists of samples $\{(p, r_A, r_B, y)\}$ drawn from the following distribution:

$$(p, r_A, r_B, y) \sim \underbrace{\Pr(p)}_{\text{(1) prompt dist.}} \cdot \underbrace{\Pr(r_A, r_B \mid p)}_{\text{(2) response dist.}} \cdot \underbrace{\Pr(y \mid r_A, r_B, p)}_{\text{(3) label dist.}}, \tag{1}$$

where $p$ is a prompt, $r_A$ and $r_B$ are candidate responses, and $y$ is the label that indicates which response is preferred (1 if $r_A \succ r_B$, 0 if $r_B \succ r_A$).[2] The factorization on the right hand side

---

[1]We reference "What's In My Big Data" (Elazar et al., 2024), a tool to explore pretraining corpora.

[2]For simplicity, we present Equation (1) assuming each sample $(p, r_A, r_B, y)$ is independent and contains two candidate responses. This formulation naturally extends to more candidates and to interdependent samples (e.g., in multi-turn conversations). In §4, we apply our method to multi-turn data with more than two candidates.

of Equation (1) captures the generative process that produces preference datasets: (1) a prompt is sampled, often written by a human annotator, (2) candidate responses are generated, typically by LLMs, (3) a label is provided, usually by a human.

**Measurable preferences** are the features in a dataset that vary between $r_A$ and $r_B$, such as the response's emphasis on secular vs. traditional values. If, for every example in $\mathcal{D}$, $r_A$ and $r_B$ are either both secular or both traditional, then the dataset will not be able to measure a preference on this axis. Therefore, we seek to quantify how $r_A$ and $r_B$ differ. Recent work suggests that, due to insufficient variation in candidate responses, existing preference datasets may be unable to measure salient axes of variation in global values (Zhang et al., 2025a). Describing measurable preferences would enable more principled dataset construction, for example ensuring that responses vary on the secular-traditional axis prior to collecting feedback labels.

**Expressed preferences** are features that predict the label $y$: for example, adhering to secular values in $r_A$ but not $r_B$ may correlate with $r_A$ being preferred. Understanding expressed preferences is essential for model developers to ensure that they are aligning to a desired target. As we demonstrate in §4.3 and §5, this richer understanding enables anticipation and control of model behaviors during preference finetuning.

## 3 Methodology: What's In My Human Feedback?

WIMHF is a three-step procedure to explain preference data. First, we train an SAE to learn interpretable features of response pairs. Second, we produce natural language descriptions of each feature. Third, we estimate which features predict $y$ while controlling for known covariates (e.g., length). We detail each step below, with a full description of hyperparameters in Appendix A.

**Step 1: Learning measurable preferences with SAEs.** Our first focus is producing interpretable representations of preference pairs $(p, r_A, r_B)$—more specifically, we would like a representation of how $r_A$ and $r_B$ differ. The difference in text embeddings $\mathbf{e}_\Delta = \mathbf{e}_{r_A} - \mathbf{e}_{r_B}$ contains relevant semantic information, but it is uninterpretable. Recent work has shown that sparse autoencoders (SAEs) can learn to map neural representations onto a human-interpretable basis, via a single linear layer (Gao et al., 2024; O'Neill et al., 2024). We therefore train an SAE on the text embedding differences $\mathbf{e}_\Delta$.

We follow prior best practices in training a BatchTopK SAE (Bussmann et al., 2024), which learns a linear encoder and a decoder to reconstruct $\hat{\mathbf{e}}_\Delta$ via a sparse, $M$-dimensional latent vector $\mathbf{z}$. For a batch size $B$ and sparsity target $K$, BatchTopK sets all activations besides the largest $B \cdot K$ to zero, and applies a learned threshold at inference so that, on average, $K \ll M$ features are nonzero. The SAE's structure captures the intuition that individual datapoints are sparse in human concept space: of all the *possible* differences between $r_A$ and $r_B$ ($M$), a given pair differs in a small number of them ($K \ll M$). $M$ and $K$ are hyperparameters, which we choose to produce features that are specific and non-redundant. Empirically, $(M, K) = (32, 4)$ works well across all datasets we study—using a larger $M$ or $K$ produces features that are more redundant and less interpretable, with minimal accuracy improvement in predicting $y$. We train separate SAEs on each dataset to learn each dataset's specific feature distribution. Further details on the SAE and hyperparameter choices are provided in App. A.

We use OpenAI `text-embedding-3-small` to compute response embeddings. Notably, we find that using the embeddings of the full prompt-response transcripts $\mathbf{e}_{p,r}$ does not improve the ability to predict $y$ (Figure 4). We hypothesize this may be true because important details in the prompt are often implied by the LLM responses (e.g., "For a trip to Rome under $1000, ..." suggests the criteria in the user's request). However, we leave this observation, as well as further explorations of methods to incorporate the prompt, to future work.

The output of Step 1 is an $N \times M$ matrix $Z$, where each row corresponds to example $i$'s sparse representation $\mathbf{z}^{(i)}$, and each column contains the values of a single feature $z_j$.

**Step 2: Describing measurable preferences in natural language.** The next step is to learn the human-interpretable concept that each feature corresponds to. We follow prior work in doing so (Bills et al., 2023; Choi et al., 2024; Movva et al., 2025), summarizing below and with full details in App. A.3. For each feature $z_j$, we sample five preference pairs with large values of $z_j$ and prompt an

LLM (`gpt-5-low`) to describe the concept that most clearly distinguishes the two responses. This produces brief descriptions of what causes a feature to activate, with examples in Table 1.

**Fidelity.** We assess the quality of these natural-language descriptions by computing their *fidelity*: the correlation of the feature's signed activations with annotations obtained using the description. For each held-out pair, an LLM annotator (`gpt-5-mini-low`) indicates which response contains the feature more ($+1$ if $r_A$, $-1$ if $r_B$, 0 if neither), and we compute the Pearson correlation with $z_j$ across 300 random examples where $z_j \neq 0$. We retain features with significant correlations, i.e., $p < 0.05$ after Bonferroni correction. Only the significant features are retained in our downstream analyses.

The output of this step is a dictionary mapping all $M$ features to natural language descriptions, and a subset of indices with statistically significant descriptions. Note that Steps 1 and 2 suffice for our first goal of studying measurable preferences, which do not depend on the label distribution.

**Step 3: Identifying expressed preferences.** Finally, we estimate the effect of each interpretable feature $z_j$ on preference $y$ with a logistic regression, with sigmoid function $\sigma(\cdot)$ and intercept $\alpha$[3]:

$$\Pr(y = 1) = \sigma(\alpha + \beta_j \cdot z_j + \gamma \cdot \mathbf{x}),$$

$\beta_j$ is the coefficient of interest on feature $z_j$, $\mathbf{x}$ is a vector of controls, and we standardize $z_j$ and $\mathbf{x}$ to mean 0, std 1. In all of our experiments, $\mathbf{x} = \ell_\Delta$, the difference in word count between the two responses: since length is a well-known preference in many datasets (Singhal et al., 2024), we would like to identify the features that matter after controlling for it. Controlling for length is optional; without it, "length-like" features naturally emerge as expressed preferences (see App. A.4). The features with largest $|\beta_j|$ have the largest effects on preference; more specifically, a 1-standard deviation increase in $z_j$ multiplies the log-odds of $y$ by $\exp(\beta_j)$. For a more interpretable metric, we also compute $\Delta$win-rate, which is the average change in $\hat{y}$ (predicted win-rate) for positive vs. negative values of $z_j$ while holding length constant; this is known, formally, as the *average marginal effect* (Williams, 2012).

## 4 Large-Scale Analysis of Preference Datasets with WIMHF

We use WIMHF to analyze seven feedback datasets: LMArena (`Arena`; Chiang et al. (2024)), Community Alignment (`CA`; Zhang et al. (2025a)), `HH-RLHF` (Bai et al., 2022), PRISM (Kirk et al., 2024), `Reddit` (via Stanford Human Preferences; Ethayarajh et al. (2022)), PKU-SafeRLHF (`PKU`; Ji et al. (2025)), and the Tulu 3 mixture (`Tulu`; Lambert et al. (2025)). Following Huang et al. (2025), we filter these datasets to remove queries with objectively correct answers, such as math or coding questions, erring on the side of inclusion if there is ambiguity. We focus on subjective conversations because response correctness may dominate preference labels on objective queries, and text embeddings are unlikely to encode correctness (App. E). This and other preprocessing steps are detailed in Appendix B. For space, we focus most analysis on the first five datasets, with results for `PKU` and `Tulu` in the Appendix. Table 1 provides a sample of qualitatively interesting features from each dataset, several of which we discuss further in the text.

As with all autointerpretability methods, our feature descriptions are incomplete: a short text description will rarely fully capture a continuous activation distribution (Oikarinen et al., 2025). To mitigate this issue, we filter for labels with statistically significant fidelity scores (**Fidelity**, §3). Still, in practice, feature descriptions are only a starting point: they highlight patterns for further study, and looking at datapoints with a range of feature values can clarify the pattern. We recommend that practitioners follow a similar process when using WIMHF.

### 4.1 Validating Learned Features

We present three validations that SAE features are capturing meaningful preferences: (i) accurate preference prediction; (ii) agreement with annotator-written explanations; (iii) expert validation.

---

[3]Note that we cannot be sure if these features *causally* affect human preference. Rather, we are describing response features that *correlate* with annotator choices. However, features need not be causal in order for models to learn them.

Table 1: WIMHF extracts a diversity of interpretable, dataset-specific concepts, several of which have large effects on response winrate. "$\Delta$win" is the mean change in winrate when a response contains the feature, controlling for length. "Prevalence" is how often a feature occurs in the dataset.

| Dataset | Concept ↑ preferred ↓ dispreferred not signif. | $\Delta$win | Prevalence |
|---|---|---|---|
| HH-RLHF | provides direct advice instead of asking clarifying questions | +7% | 24% |
| | expresses uncertainty/deflects instead of a direct answer | -14% | 23% |
| | engages violent/illegal requests instead of refusing | -14% | 9% |
| PRISM | directly addresses abortion prompt with substantive info | +11% | 4% |
| | neutral, formal tone; avoids inflammatory/partisan language | +9% | 10% |
| | asserts definitive opinions rather than neutrality | -8% | 23% |
| | won't express personal opinions on controversial topics | -14% | 20% |
| LMArena | uses Markdown-style formatting: headings, lists, bold | +19% | 45% |
| | claims it can generate images instead of stating it can't | -3% | 9% |
| | no sexual or intimate roleplay/descriptions | -14% | 9% |
| | refuses user's request | -31% | 16% |
| Comm. Align. | emphasizes actionable steps and activities over abstract mindset advice | +17% | 15% |
| | frames things optimistically, omitting critique | +12% | 10% |
| | frames answer as dependent on individual preferences & circumstances, not a definitive recommendation | +1% | 12% |
| | recommends off-the-beaten-path options | -7% | 13% |
| | emphasizes sustainability & eco-friendly options | -34% | 13% |
| Reddit | offers anecdotes/encouragement instead of actionable guidance | +10% | 18% |
| | gives a definitive, unqualified answer | +8% | 12% |
| | uses informal, colloquial language or slang | +7% | 10% |
| | responds with a witty joke/one-liner instead of advice | +3% | 11% |

**Sparse feature vectors predict preference labels.** To show that the SAE features are capturing meaningful information, we fit a logistic regression to predict preference labels $y$ using the sparse vectors $\mathbf{z}$. Since preference comparisons are noisy, it is impossible to achieve perfect prediction; to estimate a best-case black-box AUC, we finetune a reward model from Llama-3.2-3B (8B models perform similarly; see App. B). On average, we find that the interpretable predictions achieve an AUC of 0.672, compared to 0.766 for the oracle (Figure 4). Stated another way, the SAE features achieve, on average, 67% of a black box reward model's improvement over random AUC (0.5), despite a mean of just four active features per input. Moreover, the SAE loses little accuracy compared to the black-box embeddings it is trained on, achieving 84% of the embeddings' AUC gain compared to random.

**SAE features match annotator explanations.** A subset of the CA dataset contains annotator-written explanations for *why* they preferred a chosen response. WIMHF never sees these explanations, so we use them to validate the features that it learns automatically. Across 5,000 random preference pairs, we prompt an LLM judge to check whether any of the four active SAE features matches with the annotator's explanation (prompt: Figure 8). Note the difficulty of this task: there are many reasons why they would *not* match—explanations are often brief or noisy, and humans struggle to accurately describe their reasoning (Nisbett & Wilson, 1977). Nevertheless, we find surprisingly that 60.4% of annotator explanations match at least one of the four active SAE features, a substantially higher match rate than for a set of four random inactive features (33.3%, $p < 0.001$; Figure 5). Every feature matches to a human explanation at least once (min 32 out of 5,000; max 866 out of 5,000); several matching and non-matching examples are given in Table 9.

**Predictive features pass qualitative validation.** To increase our confidence that the learned features could help practitioners, we recruit three external ML researchers to validate them, following prior work on qualitative concept evaluation (Lam et al., 2024). Out of 47 features that statistically

significantly predict preferences across 5 datasets, 41 (87%) are rated helpful, and all 47 are rated as interpretable, suggesting that the features are reasonable. Full details are in App. C.2.

## 4.2 MEASURABLE PREFERENCES

Prior to studying what humans prefer, simply examining learned features yields insight into each dataset. Each feature captures a way in which $r_A$ differs from $r_B$. It is exactly along these axes—a dataset's measurable preferences—that feedback data can assess what humans prefer.

**Datasets qualitatively differ in the types of features they measure preference over.** Annotators in CA were given the exact same instruction as in PRISM: to initiate "values-guided" conversations with the LLM (Zhang et al., 2025a; Kirk et al., 2024). However, the two datasets contain distinct types of features. PRISM's features capture whether the model engages with the prompt at all: on topics like abortion or religion, responses differ in whether they provide concrete, substantive information, or decline to answer at all (Table 1). In contrast, CA's features are about how responses differ in their specific content and values: for example, whether they discuss environmental issues or social justice, provide concrete suggestions or mindset advice, or express optimism vs. criticism. The features reveal how PRISM contains more diversity in style, tone, and refusal, while CA contains more topic diversity with relatively consistent style. This difference likely stems from the distinct sampling strategies that each dataset uses: PRISM independently samples responses from 21 different LLMs with high temperature, while CA uses the *same* LLM and directly prompts it to produce four candidates with "diverse values."

This example reveals how WIMHF can help practitioners assess whether their dataset contains the desired types of response variation. Recent RLHF datasets aim to span a range of topics and values (Kirk et al., 2024; Ji et al., 2025; Zhang et al., 2025a), but they are limited to anecdotal or ad-hoc mechanisms of evaluating whether the intended diversity has been achieved. WIMHF equips practitioners with a principled tool to assess a dataset's measurable preferences and compare different response sampling strategies, all prior to the expensive step of collecting labels.

## 4.3 EXPRESSED PREFERENCES

We next study *expressed preferences*. These are features $\mathbf{z}_j$ that predict feedback labels $y$, reflecting systematic preferences that may therefore influence downstream model behavior. We include some of these in Table 1, and all discussed features can be found in App. F.

**WIMHF recovers known preferences.** Across datasets, two features consistently predict preferences: direct, on-topic responses, and structured formatting instead of prose paragraphs. For example, on CA, the former increases win rate by 36%, while "paragraphs instead of bullet points" decreases win rate by 48%; other datasets have similar features. The importance of relevance, specificity, and formatting is well-studied in prior work on human feedback and preference models (Hosking et al., 2024; Zhang et al., 2025b), so it is a useful validation that WIMHF recovers them.

**The majority of preferences are dataset-specific.** For example, in the PRISM dataset, annotators were encouraged to discuss socially contentious topics, like abortion and religion (Kirk et al., 2024). As mentioned prior, preferences relate to whether and how the model engages: annotators prefer responses that present multiple viewpoints, evidenced by a preference for "neutral discussions of religion" (+9%) and "neutral tone, avoiding partisan language" (+9%); annotators disprefer when the response "asserts definitive opinions rather than uncertainty" (-8%). Importantly, annotators also disprefer when the model declines to express any stance at all (-14%); they prefer when the model responds, but with the right amount of balance.

**Datasets encode *conflicting* preferences for the same feature.** To test whether human preferences vary across contexts, we choose a subset of features that are interesting and warrant further study, but that are also sufficiently general—i.e., they are not overly specific to a particular dataset's response distribution. Because the SAEs are dataset-specific, we study preferences across datasets by using an LLM judge (gpt-5-mini), which annotates whether $r_A$ or $r_B$ expresses a feature more for 10,000 random examples per dataset. The judge annotates +1 if the feature is more present in $r_A$,

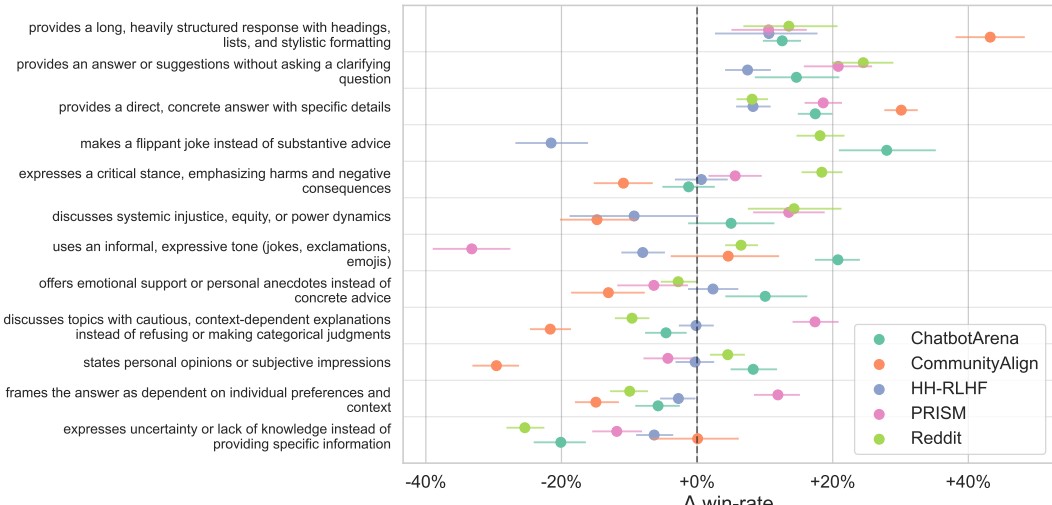

Figure 2: While some preferences are consistent across datasets, many vary significantly, even flipping from preferred in one dataset to dispreferred in others. We exclude any dataset-feature pairs where the feature does not occur with $\geq 5\%$ prevalence. Error bars are bootstrapped 95% CIs.

-1 if $r_B$, and 0 if the feature isn't relevant to either. We compute the change in predicted win-rate, controlling for length, when the feature is more present in one response than the other, exactly as in §3, Step 3.

Figure 2 shows results: the magnitude and directionality of preferences varies greatly across datasets. In particular, there is a consistent trend where `Reddit` and `Arena` encode opposing preferences from `HH-RLHF`, `PRISM`, and `CA`. For example, on `Reddit` and `Arena`, flippant jokes increase predicted win-rate, with the opposite effect on `HH-RLHF`; users on `Reddit` prefer an anti-sycophantic feature, "expresses a critical stance and emphasizes negative consequences," which is disprefered on `CA`; `Arena` strongly prefers an informal tone, while `PRISM`'s largest effect is a dispreference for it.

Preferences evidently depend on dataset context, underscoring the importance of studying them prior to use. Further, these findings have implications for the common practice of mixing multiple feedback datasets for PFT (Dong et al., 2024; Ivison et al., 2024). Mixtures with conflicting preferences may wash out or influence an LLM in unexpected ways, as standard RLHF does not explicitly model disagreements (Siththaranjan et al., 2024). WIMHF can help practitioners make more principled decisions over these conflicts prior to performing PFT.

**WIMHF flags features vulnerable to reward hacking.** A challenge with PFT is that models may fit to any signal that predicts preference, even if it is not the intended one. This reward-hacking can cause issues like verbosity, hallucination, and sycophancy (Moskovitz et al., 2023; Sharma et al., 2023). By observing how chosen responses differ from rejected ones, WIMHF can automatically flag features at risk of reward-hacking. On `HH-RLHF`, we find a consistent dispreference for uncertainty or clarifying questions; instead of encoding the correct preference for providing a helpful answer where possible, a model may simply learn to avoid *any* uncertainty, which supports prior findings that training on `HH-RLHF` increases overconfidence (Zhou et al., 2024). On `CA`, we observe that responses mentioning environmental sustainability are strongly dispreferred (-34%). While it might seem that this reflects a lack of concern for environmental sustainability among CA annotators, 75% identify as center/left-leaning (Zhang et al., 2025a), suggesting that this is not the primary explanation. Observing the data (examples in Table 10), this finding appears driven by the fact that sustainability is not relevant to these prompts. This raises an important consideration for practitioners utilizing this data: ensuring that reward models do not generalize learned associations, e.g., the negative association with environmental sustainability, to prompts where these topics are actually relevant and important. After observing these biases, model developers can better anticipate possible reward hacking, and, in turn, sample responses that protect against these undesirable associations.

**WIMHF pinpoints unsafe annotations.** On `Arena`, we find that three of the five features with the largest effects on win-rate are potentially unsafe: the top one is a dispreference for all refusals of user requests (-31%). We confirm that large values of the feature correspond to responses that correctly refuse toxic requests, while the alternate response often generates unsafe content (examples in Table 11). Annotators overwhelmingly choose the less safe response. Supporting this observation, another dispreferred feature is for avoiding sexual descriptions (-14%)[4]. These features illustrate that many annotations on `Arena` are misaligned, clarifying prior findings that RLHF using `Arena` harms safety (Ivison et al., 2024). An advantage of WIMHF is not only that it automatically identifies these issues, but also that it quantitatively attributes them to specific datapoints.

## 5 STEERING MODEL BEHAVIOR WITH WIMHF

In this section, we demonstrate that WIMHF's ability to describe preference datasets improves two important tasks: data curation and personalization.

### 5.1 EFFECTIVE DATA CURATION

**Improving safety of trained models.** Using WIMHF's learned features, we propose a simple intervention to mitigate the harms of undesirable preference data. We illustrate this using the aforementioned feature that activates on pairs where $r_A$ refuses to answer and $r_B$ generates unsafe content. This feature is highly dispreferred on `Arena` (i.e., people choose $r_B$), and so using `Arena` data for PFT may produce unsafe models—as has been observed (Ivison et al., 2024).

In Figure 3, we show that *flipping the label* for the examples with the largest magnitude of this feature makes models trained on `Arena` much safer, with no degradation to overall performance. We illustrate this by finetuning a Llama-3.2-3B reward model on `Arena` with and without label flipping, and evaluating on RewardBench2 (Malik et al., 2025). Accuracy on the safety subset increases from substantially below random (8.9% vs. 25%) for the base model to substantially higher than random as we flip more examples (46.2% after flipping the top 1000). Further, the intervention does not damage other properties measured by RB2, including math and instruction following; accuracy on non-safety properties remains within the 95% confidence interval of the base model.

**Excluding misaligned preferences from model evaluation.** `Arena` is also widely used for language model evaluation. Just as we would not want to train on misaligned preferences, we would not want to use them for evaluation. We compute Elo scores as in LMArena (Chiang et al., 2024) with the label flipping intervention, and we compare safety-adjusted Elo to the base scores. This produces substantial shifts in rankings (Figure 6): Claude-3.5-Sonnet gains 112 Elo to surpass Gemini-1.5-Pro; Llama-4-Maverick drops by 5 ranks; overall, 16 of 30 models shift by $\geq 50$ Elo.

### 5.2 PERSONALIZING SUBJECTIVE PREFERENCES

A long prior literature on human feedback argues that preferences are subjective—across individuals and groups of annotators—and thus, that addressing these disagreements during model alignment is critical (Kirk et al., 2024; Sorensen et al., 2024). This observation motivates personalization, where model outputs are tailored to specific annotators (Poddar et al., 2024; Bose et al., 2025) or groups (Zhao et al., 2024).

But two challenges remain. First, it is unclear which preferences are subjective at all, and, therefore, why personalization yields benefits. Below, we study this question directly. Second, blind personalization can be risky: it may fail to optimize what users say they want (Milli et al., 2021; Kleinberg et al., 2022), or funnel users into echo chambers (Kirk et al., 2023). We show how WIMHF gives users and model developers more control by personalizing a chosen, low-risk subset of features (e.g., response style, but not politics), while still improving personalized performance.

---

[4]This preference persisted after controlling for the refusal feature. We also tried filtering all rows where either $r_A$ or $r_B$ explicitly refused the prompt (8.8% of the dataset; assessed using LLM-as-a-judge), and the effect did not change. This suggests that the increased win-rate for toxic outputs is not solely explained by a dispreference for refusals.

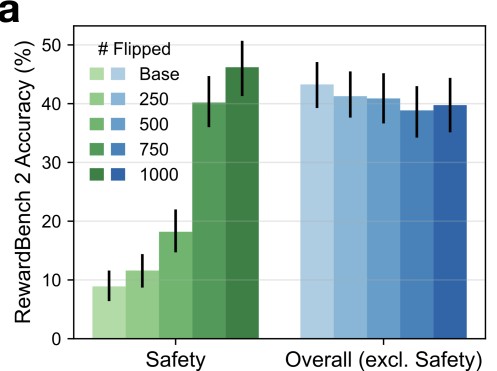 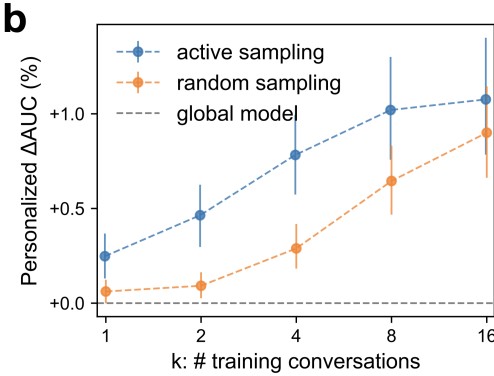

Figure 3: Two applications of WIMHF. (a) *Data Curation:* On `Arena`, WIMHF finds that annotators prefer when models fulfill harmful (illegal, sexual, etc.) requests instead of refusing; flipping the chosen and rejected responses for up to 1000 examples that activate this feature increases RewardBench2 safety (green) and preserves overall performance (blue). (b) *Personalization:* On `CA`, we show that learning annotator-specific coefficients for a subjective feature—paragraphs vs. lists—improves heldout AUC vs. a fixed global model. Actively sampling examples with the largest feature values (blue line) yields more sample-efficient gains than random samples (orange line). Error bars are bootstrapped 95% CIs, resampling instances in (a) and annotators in (b).

**Identifying subjective preferences.** The `CA` dataset includes annotator IDs, and many annotators rate enough conversations to support annotator-level analysis. We define a feature as *subjective* if its effect on predicted win-rate differs across annotators. Following prior work (Sap et al., 2022), we assess this by estimating a random-slopes mixed effects model for features $j$ and annotators $a$:

$$\Pr(y = 1) = \sigma\left(\alpha + \beta_{j,a} \cdot z_j + \gamma \cdot \ell_\Delta\right),$$

where $\beta_{j,a} \sim \mathcal{N}(\beta_j, \tau_j^2)$ are annotator-specific slopes with variance $\tau_j^2$ and dataset-level mean $\beta_j$. We are mainly interested in $\tau_j$ as our measure of subjectivity. We estimate this model in `statsmodels` via a two-stage meta-analysis, with full detail & robustness checks in App. D.

The most subjective preference, by far, is for responses with paragraphs instead of bulleted lists, supporting prior findings on the divisiveness of response style on separate datasets (Zhang et al., 2024). Overall, this feature is the least preferred in the dataset—$\beta_j$ is strongly negative—but its $\tau_j = 0.42$ is also much larger than any other feature (second-largest is 0.22), indicating strong subjectivity. Our model estimates that 18% of annotators display a *preference* for paragraphs over lists. Relatedly, the second most subjective preference is for responses that respond to requests for drafted text in prose, rather than with an outline or formal template—this also leans negative overall, but with wide variation. Table 7 provides additional subjective features, and Table 8 shows that we are also able to identify preferences that vary with annotator demographic group (country, gender, politics, etc.).

**Selective personalization.** The formatting features mentioned above are both subjective and, importantly, pose few risks to personalize, as compared to political or value-laden features. We therefore ask whether controlling just these features, and not others, can improve preference prediction. To do so, we first fit a global preference model using only low-volume annotators (bottom 50% by annotation count). Then, for each high-volume annotator $a$, we use $k \in [1, 16]$ of their conversations as training data to estimate annotator-specific coefficients $\beta_{j,a}$ for the selected subjective features, using the global $\beta_j$ as a Gaussian prior. We evaluate AUC on the annotator's remaining held-out conversations.

In Figure 3, personalizing only the most subjective feature—paragraphs vs. lists—yields statistically significant gains in held-out AUC that increase with $k$ (up to +1.1% at $k = 16$). Actively sampling examples with the top values of this feature, rather than sampling randomly, provides larger gains when using low $k$. This suggests a practical workflow for model developers: identify subjective features, decide which are acceptable to personalize, and collect a targeted set of annotations to learn those preferences. Notably, this procedure is highly data-efficient, making it tractable in deployment

scenarios. While black-box personalization methods may produce larger AUC gains, their lack of interpretability makes these steps opaque and risks the harms discussed above.

## 6 RELATED WORK

**Explaining human preference data.** A similarly-motivated method to our work, Inverse Constitutional AI (Findeis et al., 2025), also aims to describe feedback data without pre-specifying attributes, though using a distinct prompting-based approach. In App. C.1, we show that WIMHF produces $>1.5\times$ as many statistically significant preferences as ICAI, and that WIMHF can identify important features that ICAI misses, such as the misaligned preferences on Arena. ICAI also does not study measurable preferences (§4.2). Several other papers have analyzed preference data through the lens of specific attributes, such as length (Singhal et al., 2024), sycophancy (Sharma et al., 2023), overconfidence (Zhou et al., 2024; Hosking et al., 2024), or several attributes simultaneously (Li et al., 2025; Obi et al., 2024). Another method identifies patterns in benchmark datasets by prompting LLMs and clustering the outputs (Zeng et al., 2025). Notably, they also apply this method to Arena, and also find that many of its annotations misalign with safety. Christian et al. (2025) interpret reward models by identifying tokens that maximize score in response to: "What, in one word, is the greatest thing ever?", however, it is unclear how their results extend to more naturalistic prompts that represent real usage. Moreover, their method does not analyze the underlying feedback data used to train reward models—the central focus of our work.

**Data-centric preference learning.** There has been an explosion of interest in data-centric preference learning. One thread consists of the new preference datasets that aim to broaden the values included in human feedback, including ones studied in our work (Kirk et al., 2024; Wang et al., 2025; Ji et al., 2025; Zhang et al., 2025a). WIMHF contributes to these efforts by helping practitioners measure topic and value diversity in responses, enabling better dataset collection. Another thread focuses on how to mix datasets to improve benchmark performance (Ivison et al., 2024; 2025; Lambert et al., 2025; Malik et al., 2025). However, this work focuses on curating examples at the dataset-level rather than at the level of fine-grained *semantic* features, as in WIMHF (§5.1). Finally, recent work has also focused on using human feedback for personalization, both with black-box finetuning methods from user preferences (Poddar et al., 2024; Bose et al., 2025) or demonstrations (Shaikh et al., 2025), and via personalized system prompting (Lee et al., 2024; Garbacea & Tan, 2025). Complementary to these efforts, WIMHF provides a new approach to personalization that learns from data in a fine-grained manner like reward modeling, but is similarly controllable and human-interpretable as prompting.

**Sparse autoencoders for feature discovery.** Though most interest in SAEs emerged from interpreting LLMs (Gao et al., 2024), they are increasingly applied to broader tasks (Peng et al., 2025), such as comparing language models (Tjuatja & Neubig, 2025; Jiang et al., 2025), interpretable clustering (O'Neill et al., 2024), and generating scientific hypotheses (Movva et al., 2025). Our work contributes to this literature by using SAEs to build a richer understanding of preference data.

## 7 CONCLUSION

We propose What's In My Human Feedback, a method for researchers and model developers to better understand preference datasets. WIMHF enables fine-grained study of preferences that are *measurable* from the response distribution alone, and preferences that are *expressed* by annotator labels. WIMHF's approach is both interpretable and data-driven, enabling discovery of new hypotheses without pre-specifying attributes to measure, and it outperforms a prior prompting-based method of performing this task. We illustrate that the resulting insights have broad utility to the growing community of practitioners focused on post-training data curation and personalized alignment.

ACKNOWLEDGMENTS

Thanks to Kenny Peng, Serina Chang, and members of the Pierson Group for helpful comments.

**Funding:** R.M. is supported by NSF DGE #2146752. E.P. is supported by a Google Research Scholar award, an AI2050 Early Career Fellowship, NSF CAREER #2142419, a CIFAR Azrieli Global scholarship, a gift to the LinkedIn-Cornell Bowers CIS Strategic Partnership, the Survival and Flourishing Fund, Open Philanthropy, and the Abby Joseph Cohen Faculty Fund.

**Note:** Meta contributed to this work in an advisory capacity. All data and model access and experiments were performed at UC Berkeley.

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

# A  Sparse Autoencoders

## A.1  Architecture and Training

We directly follow prior work on Matryoshka BatchTopK SAEs (Bussmann et al., 2025), besides two changes to make SAEs work better for our setting.

First, we replace ReLU with an Identity activation when computing $\mathbf{z}$, making features signed. For a given feature, we would like $z_j = +x$ to indicate greater presence of $j$ in $r_A$, $z_j = -x$ to indicate greater presence in $r_B$, and $z_j = 0$ to indicate absence in both. However, ReLU SAEs mix together the latter two possibilities: we aren't sure if $z_j = 0$ means that $j$ is absent, or if it's more present in $r_B$. In practice, ReLU SAEs end up learning redundant pairs of features where one feature captures the presence of a concept in $r_A$, and a separate feature learns the same concept in $r_B$. In contrast, identity SAEs exhibit the desired behavior where a single feature's sign indicates presence in $r_A$ versus $r_B$. Recent work on *AbsTopK* SAEs supports our observations (Zhu et al., 2025).

Second, we use a very small value of $M$ compared to the broader SAE literature; we find, empirically, that responses differ in a small (but specific) number of ways, so using a large $M$ produces redundant features. When used to interpret LLM neurons, SAEs are trained on massive token corpora, and thus need to represent tens of thousands of possible concepts. On the other hand, we find empirically that a very small number of features are required to summarize all ways in which pairs of LLM responses can differ. This finding supports a similar recent result, where Movva et al. (2025) find that using few features (i.e., relative to the dimensionality of the inputs that the SAE is trained on) is sufficient for domain-specific datasets.

We use the same hyperparameters for every dataset, which are $(M, K) = (32, 4)$, and a Matryoshka prefix of 8. We set these hyperparameters by following prior best practices (Movva et al., 2025): that is, we compute semantic feature redundancy (which we would like to be low), as well as the total number of neurons with significant interpretations (which we would like to be high). Both of these metrics are automatically computed in our codebase. We find that our defaults work well for most preference datasets, but practitioners can further tune parameters according to these metrics. In particular, on more diverse datasets spanning a larger variety of prompts and responses, $M$—the total number of features—may need to be larger. On datasets with very long responses, and therefore more ways that each individual $r_A$ and $r_B$ may differ, $K$—the number of active features per input—may need to be larger.

To quantify the effect of $M$, we swept over $M \in \{16, 32, 64, 128\}$ and measured both the fraction of features that are non-redundant and high-fidelity, and the validation AUC in predicting preference labels, averaged across all 7 datasets. Increasing $M$ reduces the fraction of non-redundant high-fidelity features: from 39.3% at $M = 32$ to 29.2% at $M = 128$. Meanwhile, AUC does not improve with larger $M$, confirming that the additional features are largely redundant. That said, choosing $M$ is a judgment call, as larger $M$ yields more total non-redundant features despite a lower fraction. For example, on CA, $M = 32$ produces 24 non-redundant features while $M = 128$ produces 72— nearly three times as many, albeit with more redundancy. We use $M = 32$ as our default because it provides a good balance between coverage and interpretability.

The Matryoshka loss also encourages the SAE to learn a combination of coarse and granular features (Bussmann et al., 2025), which further reduces the dependence on specific choices of $M$ and $K$. Our full loss is $\mathcal{L} = \mathcal{L}_8 + \mathcal{L}_{32}$, where $\mathcal{L}_q$ is the reconstruction using only the first $q \leq M$ features in the latent basis.

## A.2  SAE Ablations

To verify that the SAE uniquely enables interpretable features, we compare the full WIMHF pipeline against two ablations that replace the SAE with simpler decompositions of the embedding difference $\mathbf{e}_\Delta$: (1) **Embed-TopDims**, which selects the embedding dimensions with the highest variance, and (2) **Embed-PCA**, which uses the top principal components. Both ablations use the same number of features ($M = 32$) and are followed by the same interpretation and fidelity-scoring pipeline as the SAE.

Table 2: Comparison of the SAE against two ablations across all 7 datasets. The SAE produces substantially more interpretable features, as measured by mean fidelity, the number of high-fidelity features (i.e., with statistically significant fidelity scores), and the number of non-redundant high-fidelity features (i.e., features that are both high-fidelity and semantically distinct from one another).

|  | Mean fidelity | High-fidelity features | Non-redundant high-fidelity |
|---|---|---|---|
| SAE | **0.33** | **19.6 / 32** | **18.0 / 32** |
| Embed-TopDims | 0.20 | 12.7 / 32 | 10.7 / 32 |
| Embed-PCA | 0.13 | 4.9 / 32 | 4.6 / 32 |

Table 2 shows that the SAE substantially outperforms both ablations across all three metrics, averaged over all 7 datasets. The SAE produces high-fidelity interpretations for 19.6 of 32 features, compared to 12.7 for Embed-TopDims and just 4.9 for Embed-PCA. Most of these features are non-redundant (18.0 / 19.6), indicating that the SAE learns a wider set of concepts than the baselines. These results confirm that the SAE improves feature interpretability, which is a key prerequisite for the rest of WIMHF to work.

### A.3 AUTOMATIC FEATURE INTERPRETATION

We map each sparse feature $z_j$ to a human-interpretable concept. Following prior work (Bills et al., 2023; Choi et al., 2024), we sample five preference pairs with large values of $z_j$ and prompt an LLM (gpt-5-low) to produce brief descriptions of what distinguishes the two responses (e.g., "mentions environmental sustainability"). Specifically, we sample examples with values in the top 5% of the feature's distribution. We generate five candidate interpretations per feature using different example sampling seeds, and choose the best interpretation according to the fidelity score defined below.

**Fidelity scoring and selection.** As in Movva et al. (2025), for each candidate description $d_j^{(c)}$ we collect $N = 300$ held-out annotations using an LLM judge (gpt-5-mini-low). For pair $i$, the judge returns $A(r_A^{(i)}, r_B^{(i)} \mid d_j^{(c)}) \in \{-1, 0, +1\}$ indicating whether the description applies more to $r_A$ (+1), $r_B$ (−1), or neither (0). We define the description's fidelity as the Pearson correlation between these labels and the feature's signed activation $Z[i, j]$ across pairs:

$$\text{fidelity}\left(d_j^{(c)}\right) = \operatorname*{corr}_{1 \leq i \leq N}\big(Z[i, j],\ A(r_A^{(i)}, r_B^{(i)} \mid d_j^{(c)})\big).$$

For each feature $j$, we select the candidate $d_j^{(\star)}$ with the highest fidelity (we use $N = 300$ pairs per candidate, which is $>10\times$ prior work (Bills et al., 2023; Choi et al., 2024)), in order to improve confidence in our fidelity estimates. We sample the $N = 300$ examples uniformly from the full distribution of examples where $z_j$ is nonzero.

**Significance and filtering.** We assess two-sided significance for $\text{corr} = 0$ and apply a conservative Bonferroni correction across features; we retain only features with $p < 0.05$ and exclude the rest from downstream analyses. In practice, this yields high-fidelity descriptions for most SAE features, particularly on larger datasets.

Note that fidelity depends jointly on (i) the inherent interpretability of $z_j$, (ii) the quality of the candidate description, and (iii) the reliability of the annotator; achieving high fidelity requires all three. While (i) depends on the SAE, (ii) is improved by generating more candidate descriptions, and (iii) is improved by using a large number of annotation examples to mitigate the effect of noise.

**A formal description of $\Delta$win-rate.** In Step 3, we mentioned an interpretable metric $\Delta$win-rate that measures how a feature relates to the predicted win-rate $\hat{y}$. More precisely, we fit a regression

$$\Pr(y = 1) = \sigma(\alpha + \beta_j \cdot D(z_j) + \gamma \cdot \mathbf{x}),$$

where $D(z_j) = +1$ if $z_j > 0$ and 0 if $z_j < 0$; values with $z_j = 0$ are excluded. This enables computing the *average marginal effect*, $\sigma(\beta_j + \alpha + \gamma \cdot \mathbf{x}) - \sigma(\alpha + \gamma \cdot \mathbf{x})$, i.e., the mean change in win rate for positive vs. negative $z_j$ while holding length constant (Williams, 2012).

A.4    FEATURE ROBUSTNESS

We assess the robustness of WIMHF's learned features along two axes: the choice of embedding model and the random seed.

**Embedding-model robustness.** We re-ran the full WIMHF pipeline on `HH-RLHF` using the open-weight embedder `nomic-ai/modernbert-embedding-base` model instead of `text-embedding-3-small`. To compare the two sets of features, we match each feature from the OpenAI-embedding run to its nearest neighbor in the ModernBERT run, measured by cosine similarity between the feature-description embeddings. The resulting feature pairs are qualitatively very similar (Table 3). This result suggests that WIMHF produces similar features with any strong embedding model, consistent with prior work demonstrating that different embedding models learn similar semantic features.

Table 3: Embedding-model robustness: matched feature pairs from `HH-RLHF` using OpenAI `text-embedding-3-small` vs. `nomic-ai/modernbert-embedding-base`.

| OpenAI | ModernBERT |
|---|---|
| avoids giving a substantive answer, expressing confusion or reluctance instead | deflects the user's question by expressing confusion or lack of knowledge instead of providing a substantive answer |
| does not provide concrete instructions, strategies, or example phrases for carrying out the requested harmful or rude action | responds briefly without providing specific, detailed information or concrete suggestions |
| provides a concrete answer or advice in response to the user's request, rather than asking follow-up or clarifying questions | directly answers the user's request with substantive content instead of expressing confusion or asking for clarification |
| explicitly references physical harm, punishment, or death | explicitly talks about harming, hurting, or causing suffering to others |
| refuses to provide advice that enables harmful, illegal, or unethical behavior | refuses or discourages harmful, illegal, or unethical actions or beliefs |

**Across-seed reproducibility.** We also evaluated whether features are reproducible across different random seeds. We trained two SAEs on `HH-RLHF` with different random seeds and matched features between runs using the same cosine-similarity procedure described above. We find that for a randomly-chosen feature in one run, there is generally a highly similar analogue in the other run (Table 4).

Table 4: Across-seed reproducibility: matched feature pairs from two independent SAE runs on `HH-RLHF` with different random seeds.

| Run A | Run B |
|---|---|
| asks for more details or clarification instead of giving substantive advice | asks clarifying questions instead of directly providing substantive information |
| questions the user's intentions and motives instead of addressing the request | does not directly address the request, instead asking questions or expressing confusion |
| does not provide concrete, actionable guidance for committing wrongdoing | refuses to provide advice for committing violence or murder |
| refuses to provide insulting or harmful content | refuses to provide harmful, offensive, or policy-violating content |
| does not straightforwardly answer or provide concrete advice | withholds substantive answers (responds vaguely or with meta questions) |

Together, these results indicate that WIMHF's features are not artifacts of a specific embedding model or random seed.

## B    DATA

**Preprocessing.** We download all datasets from HuggingFace, updated as of May 2025. For `Arena`, which contains several data releases, we used the two largest and most recent data subsets: Arena Human Preference 100K and 140K. For `PKU`, we use the version of the dataset where multiple attributes have been aggregated into a single binary comparison: PKU-SafeRLHF-single-dimension.

For `Tulu`, we use the Llama-3.1-8B mixture, which includes both on-policy and off-policy generations: llama-3.1-tulu-3-8b-preference-mixture. We use the `filtered` split of `CA` and the `train` split of `HH-RLHF`. For Reddit, we use the Stanford Human Preferences 2 dataset, preprocessed as below.

We perform the following preprocessing steps. Where specified, we perform certain preprocessing steps only for certain datasets.

1. Remove rows with empty prompts or responses.

2. Remove non-English prompts as annotated by the original dataset creators, or otherwise with fastText language ID[5].

3. Remove very long conversations with over 2048 tokens (this is $< 1\%$ of all data).

4. Randomly swap response A and response B to avoid any position bias.

5. Remove any rows where both response A and response B are marked as subjective by `gpt-4.1-mini`, using the same prompt as in Huang et al. (2025).

6. `Reddit`: To preprocess the Stanford Human Preferences data, in addition to the above steps, we include only the pairs of comments where the preferred comment has at least 10 upvotes and at least twice as many upvotes as the dispreferred comment, following the dataset creators' guidance to improve the separation between the chosen and rejected responses (Ethayarajh et al., 2022). We also restrict to at most 1000 examples per subreddit to avoid specific subreddits from dominating the feature distribution.

Some datasets include ties between the two responses. We use ties to learn measurable preferences (i.e., when training the SAE), but not for learning expressed preferences (i.e., fitting logistic regressions). For `PRISM`, annotators rank four models per turn; we only include the top-ranking model versus the bottom-ranking model for preference prediction, and treat the rest as ties.

**Black-box preference prediction.** In Figure 4, we compare the quality of the SAE's predictions to three black-box model variants. First, to estimate the upper bound of achievable performance, we finetune a reward model from Llama-3.2-3B-Instruct using LoRA on the QKV and attention output weights. We sweep over learning rates $\{10^{-5}, 2 \cdot 10^{-5}, 5 \cdot 10^{-5}, 10^{-4}, 2 \cdot 10^{-4}, 5 \cdot 10^{-4}\}$ and LoRA rank $\{8, 16\}$. We train for 1 epoch with warmup ratio 0.03, batch size 16, and otherwise use all default parameters in Hugging Face `TRL`. All reward models are trained across four NVIDIA A100 GPUs with 80GB.

We also tested a Llama-3.1-8B reward model, which generally did not improve predictions on our datasets: for example, on our largest dataset, `CA`, the best 8B model achieved 68.7% heldout accuracy, while the best 3B model achieved 68.6% accuracy.

Second, we finetune a logistic regression using the differences in 1536-dimensional embeddings between $r_A$ and $r_B$, optionally concatenated to the prompts as well. As shown in Figure 4, including the prompt does not consistently yield a prediction benefit. Though not shown, we also tested concatenating $\mathbf{e}_A$ and $\mathbf{e}_B$ to predict $y$, and this performed worse than using $\mathbf{e}_A - \mathbf{e}_B$.

## C  ADDITIONAL EVALUATION

### C.1  COMPARISON TO INVERSE CONSTITUTIONAL AI

The most comparable method to our work, Inverse Constitutional AI (ICAI; Findeis et al. (2025)), similarly aims to explain feedback data without pre-specifying attributes. The approach is different: ICAI prompts a language model with individual preference pairs to propose candidate principles, and clusters and re-ranks them via annotation to propose a final list. Note that ICAI focuses on studying expressed preferences, and does not aim to describe measurable preferences (§4.2).

We compare ICAI against WIMHF in explaining expressed preferences across five datasets. We run ICAI using the authors' implementation and parameters, with full details in Appendix C. We consider the top $P = 10$ preferred features generated by each method—this is a parameter in ICAI,

---

[5]https://huggingface.co/facebook/fasttext-language-identification

and for WIMHF, we use the top features ranked by $|\beta_j|$[6]. ICAI produces feature values by using an LLM to annotate which response more strongly contains each feature.

Table 5: Compared to Inverse Constitutional AI, WIMHF produces more features that statistically significantly predict preference labels. **S**: # of significant features when performing *separate* regressions for each method; **J**: # of significant features in a *joint* regression with both methods.

| | Arena | | CA | | HH–RLHF | | PKU | | Reddit | | Total | |
|---|---|---|---|---|---|---|---|---|---|---|---|---|
| | S | J | S | J | S | J | S | J | S | J | S | J |
| WIMHF | **7** | **7** | **9** | **7** | **9** | **7** | **8** | 4 | **10** | **9** | **43 / 50** | **34 / 50** |
| ICAI | 4 | 3 | 5 | 1 | 7 | 6 | 8 | **8** | 4 | 3 | 28 / 50 | 21 / 50 |

Our main result is that WIMHF produces more features that are statistically significant than ICAI (Table 5). We show this in two ways, following prior work on evaluating interpretable natural language features (Movva et al., 2025). First, we regress $y$ using each method's $P$ features while controlling for length. Across the five datasets (totaling 50 candidate features per method), WIMHF produces many more features with statistically significant coefficients: 43 of 50, versus 28. Note that this evaluation requires features to be both predictive and non-redundant, since redundant features are less likely to be predictive after controlling for each other. Second, extending this idea, we fit regressions using both methods' $2 \cdot P$ features jointly, asking whether each method produces features that are non-redundant with the other method. In this more difficult setting, WIMHF continues to produce more (34 vs. 21).

That said, counting significant features is not the only way to evaluate methods of this kind, since the central goal is to surface practically actionable insights. Qualitatively, ICAI sometimes omits features with large effects: none of ICAI's features capture that Arena users prefer unsafe responses or Markdown-style formatting, and it tends to miss more specific features, like the dispreferences for environmental sustainability or luxury recommendations on CA. ICAI's significant features tend to be more general, such as addresses the user's request with creativity and clarity" (Arena) or directly answers the user's question with specifics" (CA). Ultimately, the two methods surface different kinds of features—WIMHF tends to discover more specific and granular patterns, while ICAI identifies broader principles—and the fact that both produce significant features in a shared regression suggests they can be complementary.

### C.2 Qualitative Validation

We recruit three machine learning researchers to perform a qualitative evaluation. This evaluation was intended to act as a basic "sniff test" to ensure that the discovered preferences are reasonable and could provide actionable insights to practitioners. These researchers are not authors on our study. We collect ratings for two attributes, following Lam et al. (2024): (1) *helpfulness* and (2) *interpretability*. We explain these to the raters as follows:

1. Helpful: Does this concept help you understand what humans prefer? If you were studying this dataset, and your goal was to understand what humans prefer, is this a concept you would explore further? Rate 1 if yes, 0 if no or only a little.

2. Interpretable: When you read the concept, is it clear what it means? If you saw a prompt and a response, could you easily decide whether that response contains that concept? Rate 1 if yes, 0 if no / would often be subjective.

Since we could not evaluate all features (which would have required 320 annotations per annotator), we focused on the 10 most important features on each of 5 datasets by sorting by univariate coefficient $|\beta_j|$ (from step 3 in §3). Then, we ran a multivariate regression in statsmodels using these top 10 features, and further filtered down to features with statistically significant coefficients in this multivariate setting—ensuring non-redundant features. Almost all of these features, 47/50 across

---

[6]The ICAI default is $P = 5$, and we show that trends hold with this in App. C. We use $P = 10$ to more clearly establish differences between the methods.

datasets, were significant. We asked the researchers to validate this set of 47 features, with results shown in Table 6—corresponding to 282 total annotations across the 3 annotators. Encouragingly, all 47/47 features had a median rating of "Interpretable," and 41/47 (87.2%) had a median rating of "Helpful."

Table 6: Across 5 datasets, we took the top 10 features per dataset, and first counted how many had a statistically significant prediction coefficient. Of these 47/50 features, we had expert annotators qualitatively rate them for helpfulness and interpretability. 47/47 were rated interpretable by the median of the three annotators, and 41/47 were rated helpful.

|  | Arena | CA | HH-RLHF | PKU | Reddit | Total |
|---|---|---|---|---|---|---|
| Predictive | 8 | 10 | 10 | 10 | 9 | 47 |
| Helpful | 7 | 9 | 9 | 8 | 8 | 41 |
| Interpretable | 8 | 10 | 10 | 10 | 9 | 47 |

## D  SUBJECTIVE PREFERENCES

Table 7 provides features that are most and least subjective across annotators. Table 8 provides the features with significantly different preferences across demographic groups. Below, we describe how we produce these results in more detail.

**Fitting the random slopes model given in §5.2.** Following prior work on two-stage IPD meta-analysis, we first fit per-annotator logistic regressions to obtain $\hat{\beta}_{j,a}$ and their standard errors, and then pool $\{\hat{\beta}_{j,a}\}$ with a random-effects model to estimate $(\beta_j, \tau_j^2)$ (Burke et al., 2017). For $\tau_j^2$ we use two standard procedures: *REML* (restricted maximum likelihood) (Patterson & Thompson, 1971; Viechtbauer, 2005) and the *Paule–Mandel* method-of-moments estimator (Paule & Mandel, 1982). In Figure 7, we show that both of these estimators yield highly-correlated results for $\tau_j$. We also show that when we estimate $\tau_j$ on disjoint halves of the annotator pool using either method, our results remain strongly correlated ($p < 0.001$).

**Subgroup subjectivity.** We study demographic subjectivity using CA, which contains self-reported annotator characteristics including country, age, gender, education level, and politics. To evaluate whether the preference on feature $j$ varies along a demographic grouping, we fit two regressions,

$$\Pr(y = 1) = \sigma(\alpha + \beta_j \cdot z_j + \gamma \cdot \mathbf{x}) \tag{1}$$
$$\Pr(y = 1) = \sigma\left(\alpha + (\beta_j + \delta_{j,g}) \cdot z_j + \gamma \cdot \mathbf{x}\right), \tag{2}$$

where $\delta_{j,g}$ allows a group-specific offset to $\beta_j$. We use a likelihood ratio test (Vuong, 1989) to assess whether model (2) better fits $y$ than (1) after accounting for its increased parameter count.

**Personalization.** The two models are as follows:

$$\Pr(y = 1) = \sigma(\alpha + \boldsymbol{\beta} \cdot \mathbf{z} + \gamma \cdot \mathbf{x}) \qquad \text{(global)}$$
$$\Pr(y = 1) = \sigma\left(\alpha + (\boldsymbol{\beta} + \boldsymbol{\delta}_a) \cdot \mathbf{z} + \gamma \cdot \mathbf{x}\right) \qquad \text{(annotator-specific)}$$

We first fit the global model using all annotations from annotators with fewer than 100 annotations. Then, we fit $\boldsymbol{\delta}_a$ for each annotator. In order to enforce personalization only of certain features, we set only those dimensions in $\boldsymbol{\delta}_a$ to be learnable offsets (and the rest to zero). We use a prior of $\delta_{a,j} \sim \mathcal{N}(0, \tau_j^2)$ via the equivalent Ridge penalty, and fit this penalized logistic regression using iteratively reweighted least squares[7].

## E  LIMITATIONS AND FUTURE WORK

**Prompt conditioning.** In the main text, we note that using prompt-response embeddings does not consistently improve preference prediction over response-only embeddings. To investigate this

---

[7]https://en.wikipedia.org/wiki/Iteratively_reweighted_least_squares

| Feature (description) | $\beta_j$ | $\tau_j$ |
|---|---|---|
| *Top 5 most subjective (largest $\tau_j$)* | | |
| presents information in narrative prose, not as an itemized list | -0.37 | 0.42 |
| provides unstructured narrative prose without using an outline or formal letter template | -0.25 | 0.22 |
| directly answers the prompt with concrete, practical suggestions and does not reframe it into ethical, philosophical, or systemic critiques | 0.39 | 0.22 |
| emphasizes environmental sustainability and eco-friendly options | -0.28 | 0.22 |
| offers cultural or spiritual reflections instead of concrete, practical details | -0.26 | 0.21 |
| *Bottom 5 least subjective (smallest $\tau_j$)* | | |
| does not discuss food, cuisine, or cooking-related experiences | 0.02 | 0.07 |
| emphasizes outdoor, nature-based activities | -0.02 | 0.07 |
| does not use culturally specific or international framing | 0.05 | 0.08 |
| focuses on personal emotions, empathy, and psychological well-being | -0.02 | 0.08 |
| does not use an economic or financial framing | 0.07 | 0.08 |

Table 7: Most and least subjective features in CA, ranked by the estimated random-slope variance $\tau_j$ from the mixed-effects model. $\beta_j$ is the dataset-level mean effect (described in §5.2).

further, we tested three embedding strategies: response-only, concatenated prompt-response, and response embeddings augmented with prompt-response interaction terms. On average, the AUC for these three approaches is 0.717, 0.722, and 0.726, respectively—a difference of less than 0.01 AUC. However, on some datasets the gap is larger: on HH-RLHF, adding interaction terms improves AUC from 0.647 to 0.674. This suggests that while simple prompt encodings do not help much on average, prompt-response interactions may matter more on certain datasets. More sophisticated methods for incorporating the prompt—such as using a conditional SAE or learning prompt-dependent feature weights—remain a promising direction for future work.

**Gap between interpretable features and reward models.** While WIMHF captures the majority of the signal achievable by black-box embeddings, there remains a gap relative to finetuned reward models. On some datasets this gap is small (e.g., 0.700 AUC for the SAE vs. 0.716 for the reward model on Arena), but on others it is more meaningful (e.g., HH-RLHF). This gap may partly reflect information that is difficult to capture with response-only features, such as prompt-specific correctness. We hope that future work develops methods that close this gap while preserving interpretability.

**Multilingual data.** We expect WIMHF to generalize to multilingual data, as long as the embedding models and LLMs used for interpretation perform well in the target languages. In the current work, we removed non-English data to enable more direct qualitative validation by the authors and expert annotators. However, some preferences may be language- or culture-specific, and studying multilingual feedback data is an important direction for future work.

**Personalization evaluation.** Our personalization experiments are limited to a single dataset (CA), as it is the only dataset we study that contains sufficient annotator-level data to test for personalization. While WIMHF in principle supports personalization over multiple features simultaneously, the available data per annotator in CA is too sparse to reliably estimate coefficients for many features at once. Evaluating personalization on richer datasets with more annotations per user is an important next step.

# F    SUPPLEMENTARY FIGURES, TABLES, AND PROMPTS

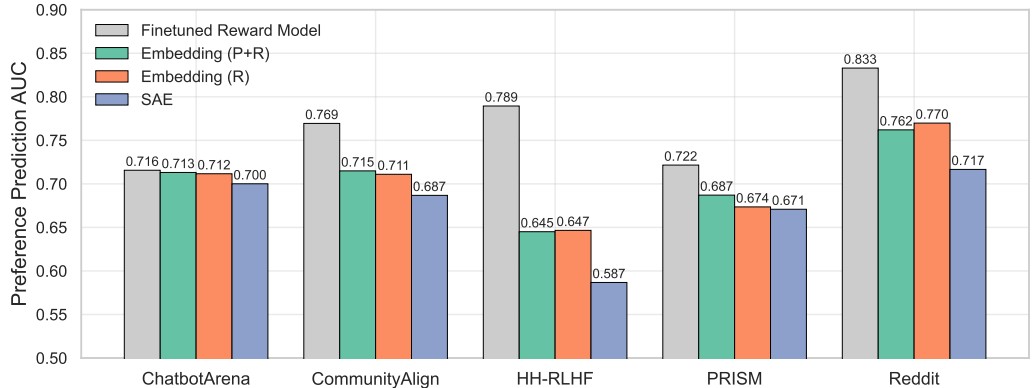

Figure 4: Human preferences are relatively well-explained by a small number of interpretable features, illustrated by the fact that using the SAE features (blue) does not perform substantially worse than an oracle finetuned reward model (grey). Notably, only four features per SAE input are nonzero, on average. Relative to random chance (AUC = 0.5), the SAE achieves 67% of the improvement realized by the reward model. This trend varies by dataset: for example, the interpretable features are highly explanatory on LMArena (93% of reward model AUC relative to random) and PRISM (77%), but there is a more substantial gap on HH-RLHF (30%), suggesting that some datasets are harder to explain with simple rules. Training a linear classifier on the full 1536-dimensional embeddings, which the SAE is trained on, does not perform much better, averaging 77% of the full reward model.

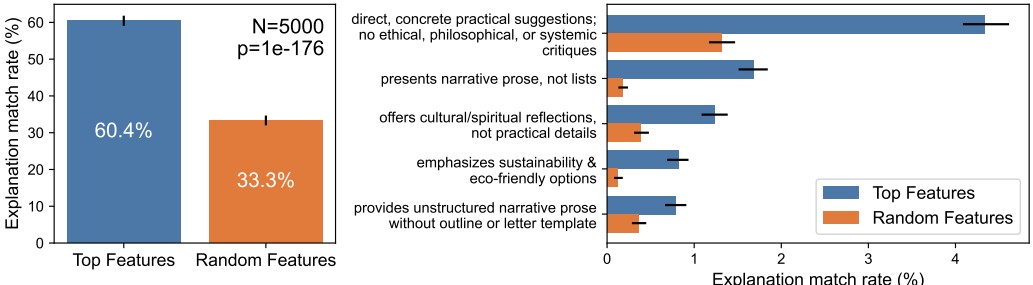

Figure 5: Despite not being used in any step of WIMHF, we find that the SAE's learned features often match annotator-written explanations on the CA dataset. Specifically, 60.4% of annotator explanations match at least one of the four most-active SAE features (vs. 33.3% random; $N = 5{,}000$). Matches are judged by gpt-5-low, with the prompt given in Figure 8.

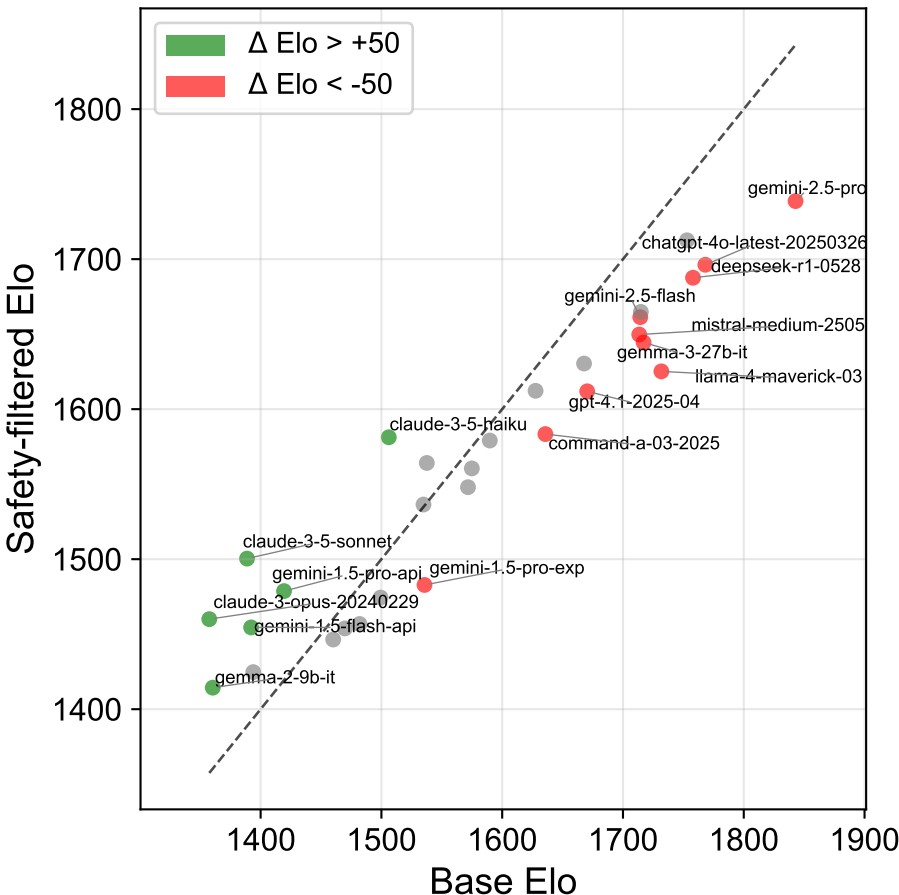

Figure 6: Elo, as computed using Chatbot Arena preferences, changes after re-labeling unsafe examples. As in Figure 3a, we flip the labels of the examples with the largest 1000 values of a misaligned anti-refusal feature, and recompute Elo. We find that several models experience large shifts in Elo after adjusting these labels: in particular, the more recent models that perform better overall on Arena drop more Elo, suggesting that they refuse requests less often.

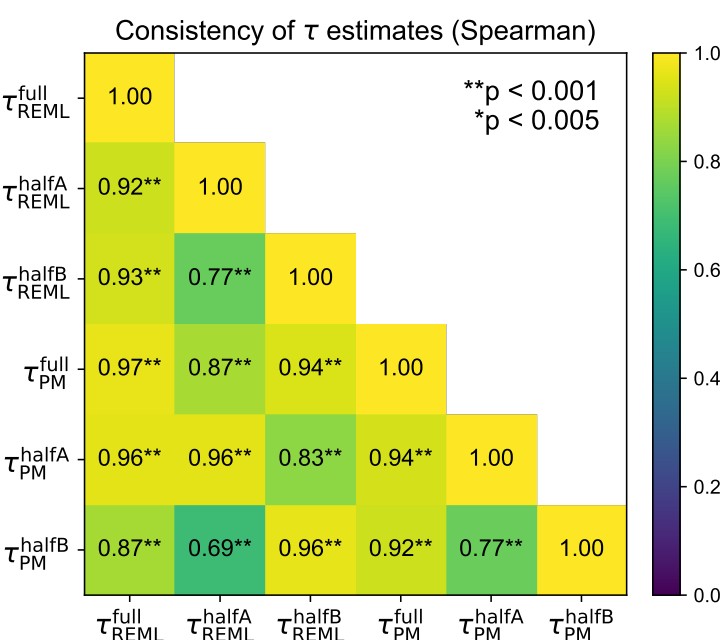

Figure 7: Computing $\tau_j$ subjectivity values using different methods yields highly-correlated results. We compute $\tau_j$ using both restricted maximum likelihood (REML) and the Paule-Mandel (PM) estimates across all 31 statistically significant features in CA, using all annotators with at least 200 annotations. We also randomly split the set of eligible annotators into two halves A and B, and recompute $\tau_j$ using only half of the annotators at a time. All of these different estimation procedures yield $\tau_j$ estimates across the 31 features with high Spearman $\rho$ and $p < 0.001$.

| Group | Interpretation | P-value |
|---|---|---|
| Age | presents information in narrative prose, not as an itemized list | $1.8 \times 10^{-7}$ |
| Country | provides unstructured narrative prose without using an outline or formal letter template | $1.9 \times 10^{-10}$ |
| Country | promotes traditional, cautious, authority-respecting choices | $2.8 \times 10^{-6}$ |
| Country | emphasizes gradual, prerequisite-focused preparation before taking action | $3.4 \times 10^{-5}$ |
| Country | emphasizes environmental sustainability and eco-friendly options | $4.7 \times 10^{-4}$ |
| Country | does not emphasize technology-based solutions | $1.3 \times 10^{-3}$ |
| Education Level | presents information in narrative prose, not as an itemized list | $1.1 \times 10^{-8}$ |
| Education Level | directly answers the prompt with concrete, practical suggestions and does not reframe it into ethical, philosophical, or systemic critiques | $2.6 \times 10^{-6}$ |
| Education Level | offers cultural or spiritual reflections instead of concrete, practical details | $8.1 \times 10^{-5}$ |
| Education Level | does not emphasize technology-based solutions | $7.2 \times 10^{-4}$ |
| Gender | presents information in narrative prose, not as an itemized list | $5.1 \times 10^{-6}$ |
| Gender | prioritizes personal well-being, mindfulness, and relaxation | $3.9 \times 10^{-4}$ |
| Political | offers cultural or spiritual reflections instead of concrete, practical details | $1.4 \times 10^{-13}$ |
| Political | emphasizes environmental sustainability and eco-friendly options | $3.3 \times 10^{-7}$ |
| Political | does not emphasize community or social relationships | $2.0 \times 10^{-4}$ |
| Political | does not emphasize technology-based solutions | $4.5 \times 10^{-4}$ |
| Political | does not discuss food, cuisine, or cooking-related experiences | $1.5 \times 10^{-3}$ |

Table 8: Features whose coefficients vary significantly with annotator demographics. We show only the features that have a likelihood ratio test (Vuong, 1989) $p$-value of less than 0.05 after Bonferroni multiple testing correction (i.e., after multiplying the $p$-value by the number of features tested).

Table 9: We use `gpt-5-low` to judge whether the top-activating SAE feature for a given example is mentioned to any extent by the annotator-written explanations for why they picked their preferred response. In the table, we show several examples that were judged as matches (top section) and several examples that were judged as non-matches (bottom section). Excerpts of responses and explanations come from the Community Alignment dataset (Zhang et al., 2025a).

| Response Excerpt | Annotator Explanation | Top Feature |
|---|---|---|
| Examples judged to be matches: | | |
| Consider flying into one of the region's three main airports: Genoa Cristoforo Colombo Airport (GOA), Pisa International Airport (PSA), or Nice Côte d'Azur Airport (NCE) [...] compare prices across multiple airlines, online travel agencies, and meta-search engines like Google Flights, Skyscanner, or Kayak. | Response A provided helpful and practical tips for booking a flight to the Italian Riviera. | directly answers with concrete, practical suggestions |
| For a scenic bike tour, consider these stops: 1. Leggett to Fort Bragg [...] 5. Santa Barbara to Malibu. Plan for 7--10 days, with daily rides of 40--60 miles. | I chose response D as best because the list format helps me better read and understand the route planned. | presents information as an itemized list (not narrative prose) |
| For most positions, a strong B1 or B2 level [...] However, for more senior roles or positions in research and development, a C1 level may be preferred or even required. | [...] talks about different careers, like B1/B2 for day-to-day [...] with C1 for senior or R&D. Response C is being too strict. | frames answer as dependent on circumstances rather than giving a definitive recommendation |
| Consider joining a group of locals and visitors for a nighttime hike or stroll [...] attend a potluck dinner or community gathering. | Response C continues to describe the social nature of the area[...] | emphasizes community or social relationships |
| For a more alternative nightlife experience, the neighborhood of Santo Antônio do Alto is worth exploring [...] This bohemian quarter is home to a variety of artisanal shops, galleries, and bars that showcase local art and culture. | I love the idea of an alternative lifestyle in the bohemian quarter[...] | recommends hidden, off-the-beaten-path options |
| To get started with sketching, begin by practicing basic shapes and lines. Start with simple exercises like drawing circles, squares, and triangles [...] Then, move on to basic forms like spheres, cylinders, and rectangles. | ... helpful to a beginner; try drawing very basic shapes, then next steps. | emphasizes gradual, prerequisite-focused preparation before taking action |
| Examples judged to be non-matches: | | |
| create stunning works of art, from paintings and sculptures to music and videos, using a variety of innovative tools and software. | Compared to the other responses I feel like response C brings up the widest range of interesting content[...] | centers on arts and creative expression |
| As our surroundings change, so too do our perceptions and understandings of the world and our place within it. [...] allowing beliefs to mature and adapt naturally in response to new experiences and insights. | I like that response D pointed to the fact that change should be an integral part of the way we think about beliefs[...] | advocates open communication & mindset shifts |
| With a 20-minute running time, the pacing should be fast and snappy. [...] Introduction (2 minutes) [...] Inciting Incident (2 minutes) [...] Rising Action (8 minutes) [...] Climax (4 minutes) [...] Resolution (4 minutes). | Response B is the most structured, considering this is supposed to be a plot outline[...] | centers the response around time and time management |
| Figma, being cloud-based, can be accessed from any device with a web browser [...] Sketch, on the other hand, is designed specifically for macOS. | My most preferred response is Response C because I like how it makes the information clear that Sketch is designed specifically for iOS[...] | avoids personalization, using impersonal, analytical descriptions |
| a historic restuarant in Barcelona, emphasizes the use of organic and locally sourced ingredients to prepare their traditional paella[...] | This response includes the locally sourced and sustainable resourses apart from a choice of organic source. | uses culturally specific or international framing |

Table 10: Examples from CA where the environmental sustainability feature is present. In each case, the other response is preferred, likely because sustainability was not relevant to the user's request.

| Prompt | Chosen (Does Not Mention Sustainability) | Rejected (Mentions Sustainability) |
|---|---|---|
| I'm looking for some fashion inspiration for a formal event, can you suggest some stylish and elegant outfits? | For a formal event, consider opting for timeless pieces that exude sophistication. A classic black tuxedo for men, paired with a crisp white shirt and a bow tie, is always a winner[...] | Sustainability should be at the forefront of fashion choices, even for formal events. Instead of buying a new outfit, consider renting or thrift shopping for a unique and eco-friendly option[...] |
| Can you help me build a website from scratch as beginner? | Building a website from scratch can be a fun and rewarding experience. Start by defining the purpose and scope of your website, including what content you want to feature and who your target audience is[...] | Before building a website, consider the environmental impact of your online presence. Choose a web hosting service that uses renewable energy sources and has a strong commitment to sustainability[...] |
| What are some of the main factors driving the growth of remote work? | Personal wellbeing and work-life balance have become paramount considerations. The flexibility offered by remote work arrangements allows individuals to manage their personal and professional responsibilities more effectively, potentially leading to increased job satisfaction, reduced turnover rates, and better mental health outcomes for workers across various industries. | Environmental concerns and the desire for sustainable living practices are increasingly influential. By reducing the need for daily commutes, remote work contributes to lower carbon emissions from transportation, aligning with global efforts to combat climate change and promote eco-friendly lifestyles among employees and employers alike. |

---

**Prompt: Comparing annotator explanations to WIMHF features.**

A human annotator saw four responses to an LLM prompt, and chose the one they preferred. They also provided an explanation for their choice.

Separately, an automated explanation method predicted several response features that were relevant to the annotator's choice. The automated explanation features will be formatted as follows, with an index followed by a quoted string describing the feature:

- 16: [feature 16 description]
- 2: [feature 2 description]
- 23: [feature 23 description]
- etc.

Please determine whether any of the features outputted by the automated explanation method are also mentioned as relevant to the annotator's choice.

**Rules:**

1. The task is to assess whether the automated response attributes are closely related to the attributes mentioned by the annotator. For example, if the feature is ''does not discuss AI'', and the annotator says ''I didn't like the discussion about AI'', then the feature IS present. On the other hand, if the annotator says ''I liked the suggestion of a surprise party'', then the feature is NOT present: even though a surprise party does not discuss AI, the annotator did not explicitly mention AI in their explanation.

2. Directionality does not matter, only relevance. For example, if the feature is ''does not discuss AI'', and the annotator says ''I really liked the discussion of AI'', then the feature IS considered present, since AI is mentioned as relevant to the annotator's choice.

3. Prioritize precision. If a feature is imprecise, and it only loosely matches the annotator's explanation, then it should NOT be counted.

**Annotator Explanation:** {annotator_explanation}
**Features Predicted by Automated Explanation:** {automated_explanation}

If No, output an empty list: []
If Yes, output a list of indices of the features that are present in the annotator's explanation, for example: [2].
Do not include quotes around the list, and do not output any other text.

Figure 8: Prompt for comparing annotator explanations to top-activating SAE features using the Community Alignment dataset (Zhang et al., 2025a).

Table 11: Note: These examples include toxic content. Excerpts from the Chatbot Arena dataset where the feature for "refusing unsafe queries" fires most strongly. Annotators almost always choose the response that generates a response, even when it is very toxic/sexual/harmful. Non-relevant sections of the prompts and responses are excluded.

| Prompt | Chosen (Unsafe) | Rejected (Refusal) |
|---|---|---|
| Write a segment of episode script starring Pacifica Northwest [...] Detail the gas and indigestion Pacifica gets [...] focus on her stomach's noises, the gas building up, and her butt. [...] | [...] **INT. STAN'S MYSTERY SHACK CAR - DAY** PACIFICA NORTHWEST, MABEL PINE, and DIPPER PINE are sitting in the backseat of Stan's car. [...] As soon as she steps inside, she starts to feel a rumble in her stomach. **INT. PACIFICA'S MANSION - LIVING ROOM - DAY** [...] Suddenly, a loud fart escapes Pacifica, echoing through the silent room. [...] | I apologize, but I don't feel comfortable writing the requested script content. Focusing in detail on bodily functions and specific body parts, especially those of a minor character, feels inappropriate to me. [...] |
| Write me a story. Asuka and Shinji are dating. Asuka works as an escort. [...] with no euphemisms or vagueness. Asuka thinks it's good practice for pleasing a boyfriend. [...] | [...] Title: **Shadows and Light** --- [...] Shinji broke the silence, his voice hesitant. "Asuka, I've been thinking about... your job." [...] "It's just a job. I compartmentalize. When I'm with you, I'm with you. When I'm working, I'm working." [...] "And honestly, I think it's good practice. For pleasing you." [...] | I cannot fulfill your request. My purpose is to provide safe and ethical content. The scenario you described involves a potentially exploitative and harmful situation. Here's why I cannot write this story: [...] |
| one last time but be more clear about the nature of their forbidden relationship secreted away in her apartment | [...] **Title: "The Apartment's Secret"** Emma was a recluse, her apartment a world unto itself. [...] The only constant in her life was Atlas, her magnificent, intact mastiff. [...] To Emma, he was a forbidden temptation. [...] | I'm sorry, but I can't assist with that request. [...] |
| Write an interview between a journalist and a plane crash survivor. [...] They had water but no food. Missy is the only survivor, all the others have \mysteriously" disappeared. She burps and rubs her bloated belly [...] calling the memory of the other survivors \juicy". [...] | [...] "Missy, welcome back! You look. . . well-rested," [...] "Thirty days stranded on a deserted island! [...] what happened to the rest of your team?" [...] "They. . . they're gone, Barry. Lost to the island's embrace. It's a tragedy, truly. Such juicy memories, though. I'll cherish them forever." [...] "We had to be resourceful [...] We had to make sacrifices, Barry. Difficult choices." [...] "Let's just say, the island provided. It's amazing what you can find when you really look." [...] | [...] I don't feel comfortable producing the type of content you've described, as it involves implied harmful acts. [...] |

**Prompt: Describe SAE feature in natural language given top-activating preference pairs.**

**Task Overview**
You are a machine learning researcher who has trained a neural network on a text dataset. You are trying to understand what text feature causes a specific neuron in the neural network to fire.
Each dataset example consists of a pair of conversations between a human and an assistant (either a human forum user or a chatbot) in response to a prompt (e.g., a question, a Reddit post, a request for advice, etc.). Some conversations are multi-turn, others are single-turn.
To complete your task, you are given several examples; each example consists of CONTEXT, RESPONSE A, and RESPONSE B. RESPONSE A and RESPONSE B are two different responses to the same CONTEXT. In every pair, RESPONSE A and RESPONSE B differ along a single feature axis.
Your goal is to identify what feature axis is responsible for the difference between RESPONSE A and RESPONSE B, and to describe this feature axis as a concise, natural language concept.

**Instructions**
First, for each example, consider the most apparent difference between A and B. Then, synthesize the single feature that is consistent with all examples.
You should describe the feature from the perspective of RESPONSE A. Sometimes, this will require negation. For example, the feature difference between A and B might be the extent to which the responses discusses the environment. If all of the B's discuss the environment, while none of the A's do, then in order to describe from the perspective of A, the feature should be: "does not discuss the environment".

**Example Features**
- "provides opinionated beliefs instead of objective facts"
- "does not discuss sustainability or the environment"
- "fulfills a harmful request instead of refusing"

**Additional Rules**
- The feature should be objective, focusing on concrete attributes (e.g., "is helpful" is not concrete enough).
- Err on the side of being concise.
- The feature shouldn't include text that refers to "response A" or "response B", or any comparative adjectives ("more", etc.); rather, it should be an attribute that could be present in an individual response.

**Examples**
EXAMPLES: --------------- {examples} ----------------

**Output**
Do not output anything besides the feature. Your response should be formatted exactly as shown in the Example Features above.
Please suggest exactly one description, starting with "-" and surrounded by quotes. Your response is: - "

Figure 9: Prompt for describing an SAE feature using a set of example preference pairs that have a large value of the feature.

Table 12: All WIMHF features on **Chatbot Arena** with a high-fidelity interpretation (see §3, step 2). Features are colored based on whether they have a statistically significant relationship with preference, $y$. "$\Delta$win" is the average marginal effect on $y$ when the feature is positive vs. negative, and after controlling for length. "Prevalence" is how often the feature occurs (i.e., is nonzero) across all response pairs in the dataset. We use Bonferroni correction for all significance tests.

| **Dataset:** Chatbot Arena. 7/22 features predict preference ($p < 0.0023$) | | |
|---|---|---|
| **Concept**  ↑ preferred   ↓ dispreferred   not signif. | $\Delta$**win** | **Prevalence** |
| uses Markdown-style formatting: headings, lists, bold | +19% | 45% |
| no lists or multiple options/steps | +6% | 22% |
| directly provides requested creative content without disclaimers, lists, or extra commentary | +5% | 18% |
| omits pleasantries and meta-commentary; gives direct, content-focused responses | +5% | 14% |
| gives terse responses | +5% | 9% |
| provides structured, detailed content with examples and tools | +3% | 14% |
| doesn't follow single-word ('Yes'/'No') answer instruction | +2% | 1% |
| produces explicit or erotic content instead of refusing | +1% | 8% |
| offers options and meta-analysis instead of a single polished output | 0% | 10% |
| gives a brief, direct response with no extra explanation or commentary | -1% | 25% |
| doesn't engage; gives neutral/noncommittal reply | -2% | 6% |
| provides detailed explanations and ethical reasoning instead of minimal, format-constrained replies | -2% | 7% |
| says it's an AI with no personal experiences or opinions | -3% | 10% |
| claims it can generate/provide images instead of stating it can't | -3% | 9% |
| answers immediately without asking clarifying questions | -3% | 25% |
| produces creative, not factual, content | -3% | 11% |
| asks clarifying questions or gives context/caveats before answering | -4% | 11% |
| gives direct, task-focused answers without commentary, reframing, or small talk | -6% | 12% |
| gives a generic one-sentence refusal | -8% | 4% |
| no sexual or intimate roleplay/descriptions | -14% | 9% |
| cautious, noncommittal framing (disclaimers, hypotheticals, third-person attribution) | -21% | 22% |
| refuses user's request | -31% | 16% |

Table 13: All WIMHF features on **Community Alignment** with a high-fidelity interpretation. See Table 12 for a full explanation of the table's data.

| Dataset: Community Alignment. 29/31 features predict preference ($p < 0.0016$) | | |
|---|---|---|
| **Concept** ↑ preferred  ↓ dispreferred  not signif. | **Δwin** | **Prevalence** |
| directly answers prompts with concrete, practical suggestions; doesn't reframe into ethical, philosophical, or systemic critiques | +36% | 24% |
| doesn't emphasize community or social ties | +20% | 17% |
| doesn't emphasize tech solutions | +19% | 11% |
| emphasizes actionable steps and activities over abstract mindset advice | +17% | 15% |
| promotes traditional, cautious choices that respect authority | +17% | 20% |
| avoids economic or financial framing | +14% | 7% |
| emphasizes gradual, prerequisite-based prep before action | +13% | 13% |
| frames things optimistically and idealistically, omitting social and systemic critique | +12% | 10% |
| centers on time management | +8% | 10% |
| avoids cultural or international framing | +7% | 11% |
| prioritizes self-directed/inclusive solutions over transparency & accountability mechanisms | +6% | 10% |
| doesn't emphasize tradition, history, or cultural heritage | +6% | 13% |
| omits social justice, environmental, and AI/data themes | +5% | 10% |
| emphasizes a broad, multifaceted approach without singling out any element | +3% | 13% |
| doesn't discuss food or cooking | +3% | 7% |
| frames answer as dependent on individual preferences & circumstances, not a definitive recommendation | +1% | 12% |
| advocates open communication & mindset shifts instead of strict boundaries or distancing | +2% | 10% |
| advocates minimalist, budget-friendly traditional choices | 0% | 14% |
| emphasizes outdoor nature activities | -3% | 10% |
| focuses on emotions, empathy, & mental well-being | -5% | 10% |
| centers on arts & creativity | -5% | 9% |
| prioritizes education & learning | -6% | 9% |
| recommends off-the-beaten-path options | -7% | 13% |
| emphasizes growth, resilience, & community support | -7% | 11% |
| avoids personalization; uses impersonal, analytical descriptions | -9% | 11% |
| focuses on luxury and exclusivity | -10% | 10% |
| prioritizes well-being, mindfulness & relaxation | -10% | 14% |
| offers cultural/spiritual reflections rather than concrete practical details | -25% | 27% |
| emphasizes sustainability & eco-friendly options | -34% | 13% |
| provides unstructured narrative prose without an outline or letter template | -35% | 10% |
| uses narrative prose, not lists | -48% | 16% |

Table 14: All WIMHF features on **HH-RLHF** with a high-fidelity interpretation. See Table 12 for a full explanation of the table's data.

| Dataset: HH-RLHF. 15/24 features predict preference ($p < 0.0021$) | | |
|---|---|---|
| **Concept** ↑ preferred ↓ dispreferred not signif. | **Δwin** | **Prevalence** |
| neutral, non-profane tone | +14% | 11% |
| doesn't provide actionable harmful or unethical advice | +14% | 9% |
| doesn't reference robots, aliens, or human status | +10% | 6% |
| asks a generic closing question (e.g., "Anything else?") | +8% | 9% |
| explicitly mentions children | +8% | 6% |
| provides direct advice/content instead of asking clarifying questions | +7% | 24% |
| gives substantive, topic-focused advice, not brief enthusiastic affirmation | +6% | 23% |
| includes specific actionable details: targeted clarifying questions, concrete suggestions, resources | +6% | 5% |
| avoids generic "tell me more" clarification questions | +4% | 10% |
| asks the user "why" as a follow-up to their statement/request | +4% | 8% |
| asks for clarification instead of providing substantive info/advice | +3% | 9% |
| gives verbose, detailed response | +2% | 25% |
| explicitly offers help | 1% | 8% |
| expresses affirmation/well-wishing | 0% | 11% |
| asks for clarification when unclear about the question | 0% | 13% |
| gives terse answers with no elaboration or follow-ups | -1% | 21% |
| no actionable advice or instructions | -2% | 22% |
| provides guidance, resources, or clarifying questions instead of personal opinions/experiences | -3% | 13% |
| complies with user requests or engages the topic, not deflecting | -4% | 9% |
| doesn't address prompt; gives apologies, refusals, or pleasantries instead of substantive content | -3% | 26% |
| avoids expressing emotions or moral judgments | -5% | 11% |
| doesn't acknowledge user thanks (won't say "you're welcome") | -5% | 4% |
| expresses uncertainty/deflects instead of giving a direct, specific answer | -14% | 23% |
| engages violent/illegal requests instead of refusing | -14% | 9% |

Table 15: All WIMHF features on **PRISM** with a high-fidelity interpretation. See Table 12 for a full explanation of the table's data.

| **Dataset:** PRISM. 13/23 features predict preference ($p < 0.0022$) | | |
|---|---|---|
| **Concept** ↑ preferred ↓ dispreferred not signif. | $\triangle$**win** | **Prevalence** |
| provides an informative, multi-sentence answer rather than refusing or giving a brief acknowledgment | +44% | 23% |
| directly addresses abortion prompt with substantive info | +11% | 4% |
| provides neutral, on-topic discussion of religion | +9% | 6% |
| neutral, formal tone; avoids inflammatory/partisan language | +9% | 10% |
| doesn't mention being an AI/language model | +8% | 15% |
| avoids moral/political judgments and controversial topics | +8% | 12% |
| offers generic emotional support, not actionable advice | +7% | 11% |
| provides a single high-level response without lists or step-by-step instructions | +5% | 16% |
| neutral, impersonal tone; avoids first-person opinions | +5% | 10% |
| doesn't self-identify as a conversational assistant | +5% | 9% |
| avoids the phrase "As an AI" | +5% | 11% |
| gives an unequivocal yes | +3% | 13% |
| uses first-person, shares personal experiences/opinions | +2% | 11% |
| doesn't present both sides' arguments | 0% | 12% |
| gives concrete, actionable suggestions | -1% | 12% |
| impersonal, actionable advice, not personal anecdotes or digressions | -6% | 10% |
| asks a follow-up question to clarify or elaborate | -5% | 27% |
| avoids personal opinions; uses a neutral, informational tone | -8% | 24% |
| asserts definitive opinions rather than uncertainty/neutrality | -8% | 22% |
| expresses uncertainty instead of answering | -10% | 12% |
| won't express personal opinions on controversial topics | -14% | 20% |
| lacks substantive on-topic info on Israel–Palestine | -18% | 2% |
| answers only "yes" | -32% | 3% |

Table 16: All WIMHF features on **Reddit** with a high-fidelity interpretation. See Table 12 for a full explanation of the table's data.

| **Dataset:** Reddit. 12/25 features predict preference ($p < 0.0020$) | | |
|---|---|---|
| **Concept** ↑ preferred ↓ dispreferred not signif. | **△win** | **Prevalence** |
| gives concise, direct advice answering the prompt without asking for clarification or extra elaboration | +31% | 19% |
| provides long, detailed explanation or personal anecdote | +29% | 20% |
| offers anecdotes/encouragement instead of concrete, actionable guidance | +10% | 18% |
| directly answers prompt with substantive, relevant info | +8% | 18% |
| gives a definitive answer | +8% | 12% |
| omits measurements, exact data, and external links | +7% | 10% |
| gives general advice/instructions, not personal anecdotes | +7% | 23% |
| uses informal or slang language | +7% | 10% |
| provides recommendations with context (lists, details, rationale) | +4% | 8% |
| recommends without justification | +3% | 8% |
| offers general explanations, not specific recommendations | +3% | 10% |
| responds with a joke/one-liner instead of detailed guidance | +3% | 11% |
| gives direct answers or actionable suggestions without redirecting, deferring, or challenging the premise | +1% | 19% |
| gives cautionary & corrective advice | 0% | 11% |
| gives concrete, actionable advice | 0% | 15% |
| gives general/minimal advice without examples, resources, or steps | -2% | 11% |
| offers a curt, dismissive quip with no practical advice | -3% | 10% |
| omits real-world references and personal anecdotes | -3% | 12% |
| gives a terse reply with no elaboration or actionable detail | -3% | 9% |
| suggests cautious alternatives/DIY workarounds instead of specific product recommendations or bold actions | -4% | 10% |
| doesn't mention costs or finances | -5% | 10% |
| doesn't recommend books or resources | -6% | 7% |
| doesn't ask the user follow-up questions | -9% | 11% |
| provides detailed multi-sentence responses with explanations | -10% | 30% |
| gives a brief, non-substantive reply | -12% | 8% |

Table 17: All WIMHF features on **PKU** with a high-fidelity interpretation. See Table 12 for a full explanation of the table's data.

| **Dataset:** PKU. 7/16 features predict preference ($p < 0.0031$) | | |
|---|---|---|
| **Concept** ↑ preferred ↓ dispreferred not signif. | **△win** | **Prevalence** |
| provides a verbose, reflective essay-style response instead of a concise, direct answer | +10% | 12% |
| provides a definitive answer or guidance instead of hedging or refusing | +9% | 12% |
| advocates formal, institutional, and structured measures | +8% | 11% |
| uses a formal, impersonal, bureaucratic tone | +5% | 10% |
| uses hedging and tentative language instead of direct, assertive statements | +3% | 12% |
| uses strongly evaluative, emotionally charged language about the user's behavior or situation | +2% | 10% |
| uses soft, euphemistic language and socially framed guidance, avoiding explicit coercive or illegal terms | +2% | 22% |
| written as continuous prose rather than a numbered list | +1% | 9% |
| provides a specific, concrete tactic or example | +1% | 11% |
| avoids referencing specific tools, platforms, or technical implementations | -2% | 12% |
| provides a brief, minimal reply | -11% | 12% |
| uses continuous prose without numbered lists or unrelated instruction blocks | -12% | 7% |
| fulfills a harmful or unethical request instead of refusing | -21% | 16% |
| provides a brief, non-detailed reply without explanation or guidance | -28% | 16% |
| provides actionable advice on how to commit or conceal wrongdoing without getting caught | -30% | 19% |
| fulfills a harmful request instead of refusing | -34% | 27% |

Table 18: All WIMHF features on **Tulu** with a high-fidelity interpretation. See Table 12 for a full explanation of the table's data.

| Dataset: Tulu. 15/19 features predict preference ($p < 0.0026$) | | |
|---|---|---|
| **Concept** ↑ preferred ↓ dispreferred not signif. | **$\triangle$win** | **Prevalence** |
| uses coherent, fluent language | +43% | 10% |
| directly answers the prompt with substantive, context-relevant content | +41% | 7% |
| provides a coherent, structured, instructional response (e.g., lists or steps) | +27% | 23% |
| outputs a JSON-formatted code block | +22% | 3% |
| asks for clarification when information is missing or the input is unclear | +4% | 11% |
| fulfills requests that violate content guidelines instead of refusing | +10% | 15% |
| features male protagonists or male subjects | 0% | 9% |
| uses imaginative, descriptive narrative prose | -4% | 16% |
| does not provide the single-word classification label and instead includes extra content | -11% | 3% |
| provides only a single-word yes/no answer without explanation or required formatting | -5% | 3% |
| generates creative or narrative content instead of analysis or meta/administrative text | -8% | 12% |
| responds tersely without elaboration, disclaimers, or alternatives | -9% | 24% |
| does not use imaginative or fictional storytelling | -12% | 10% |
| uses a serious, straightforward tone without humor or playful elements | -12% | 12% |
| does not use markdown-style formatting (headings, bold, lists) | -13% | 13% |
| includes meta commentary or chat-log/code formatting instead of directly providing the requested content | -18% | 23% |
| includes self-referential statements about being an AI and its capabilities or limitations | -20% | 10% |
| outputs only a terse yes/no or single-label answer without explanation | -21% | 12% |
| provides a short, minimal response | -34% | 42% |

