# OpenReview forum: "What's In My Human Feedback? Learning Interpretable Descriptions of Preference Data"
_ICLR.cc/2026/Conference — ICLR 2026 Oral_

### Official Review · Reviewer_AVFK · 2025-10-22

**Soundness:** 3
**Presentation:** 3
**Contribution:** 3
**Rating:** 6
**Confidence:** 3

**Summary:**

The paper introduces WIMHF, a method that uses sparse autoencoders to turn human preference data into a small set of clear, natural-language features. These features explain much of the preference signal and reveal both what a dataset can measure and what annotators actually prefer, surfacing harmful biases like preferring non-refusals in Chatbot Arena. The authors show practical gains: flipping labels on unsafe Arena pairs markedly improves safety without hurting overall performance, and the features act as interpretable knobs for controlled personalization. Overall, WIMHF offers a simple, data-driven way to audit and steer preference datasets for alignment.

**Strengths:**

1. The paper learns sparse, natural-language features from preference pairs using a sparse autoencoder, so it can explain both what a dataset can measure and what annotators actually prefer. These features capture a large share of the preference signal (about two-thirds of a black-box model’s gain), align with annotator explanations, and beat a prior baseline.

2. WIMHF is applied to seven widely used feedback datasets, revealing which preferences are measurable and which are realized. It also shows how data collection choices affect measurable preferences, e.g., diversity differences between multi-model high-temperature sampling and prompting a single model.

3. The method leads to actionable safety improvements. IT flags unsafe preferences in Chatbot Arena, such as judges disfavoring refusals and favoring sexual/toxic outputs. Flipping labels on the most affected pairs boosts RewardBench2 safety accuracy from 8.9% to 46.2% without hurting non-safety performance, turning the analysis into a simple, effective curation step.

4. The paper identifies subjective features at the annotator level and shows that learning annotator-specific weights improves held-out AUC. It further demonstrates that actively sampling high-value feature examples is more sample-efficient than random sampling, giving a simple recipe for controlled personalization.

**Weaknesses:**

1. WIMHF uses LLMs to write feature descriptions and to judge “fidelity,” and also relies on an LLM judge for cross-dataset checks. This may introduce circularity and judge bias. Consider adding human audits, multiple independent judges, and agreement tests to stress-test these steps.

2. The method drops prompts because including them did not help simple prediction, but many preferences are conditional on the prompt. This can mislead downstream use. Maybe add prompt-aware features, counterfactual tests, and prompt-conditioned regressions to separate context-specific from global preferences.

3. Interpretable features recover only ~⅔ of a reward model’s AUC gain on average, and much less on HH-RLHF, suggesting important signals are still missing.

4. The label-flipping demo uses one small reward model and one main safety benchmark, then infers large leaderboard shifts. It would be better to validate across model scales/family and include one more safety benchmark.

5. The pipeline removes non-English/very long items and even drops rows where both responses are marked “subjective” by an LLM. These choices could suppress hard cases and alter subjectivity estimates. Please report ablations on each filter (especially the subjective filter) and consider including multilingual analyses to test robustness.

6. Personalization is promising but thinly evaluated. Gains are modest and shown mainly on one dominant feature for high-volume annotators. Extend to multi-feature personalization, low-data users would be better.

**Questions:**

1. How robust are the learned features to the choice of LLM used for description and fidelity checks?

2. In the evaluation and Table 1, why don’t the colors (preferred vs. dispreferred) align with the “∆win” values? Do the colors indicate statistical significance? This could be made clearer.

---

> ### Author Response · Authors · 2025-11-20
> **Author Response to AVFK (1/2)**
>
> ### Overview
>
> Thank you for your positive review! We’re glad that you observed many strengths of our method: it “reveal[s] both what a dataset can measure and what annotators actually prefer”, and it demonstrates practical value for safety and personalization.
>
> We appreciate your questions and suggestions, and we performed multiple new experiments to address them, described below. Our experiments include (1) a new human audit of the feature descriptions, (2) a new experiment testing other ways of encoding the prompt, and (3) extending our reward model analysis in Sec 5.1 to two new models and a new reward modeling benchmark. We believe these experiments help address all of your concerns.
>
> ### Responses
>
> *WIMHF uses LLMs to write feature descriptions and to judge “fidelity,” [...] Consider adding human audits, multiple independent judges, and agreement tests to stress-test these steps.*
>
> Good point! We added a new human audit to validate whether our feature descriptions are accurate, and whether fidelity is a good measure of accuracy.
>
> We validated 25 features with the highest fidelity and 25 features with the lowest fidelity. We randomly shuffled the features, and then for each feature, we manually compared its feature description against the five (prompt, response A, response B) triplets that were used to generate its description. Using this data, we checked whether the feature descriptions were accurate.
>
> We had both a human and GPT-5.1 perform this analysis, with similar results. See results below:
>
> | Judge           | Top-25 Fidelity | Bottom-25 Fidelity |
>   | ---                | ---    | ---       |
>   | Human              | 25/25   | 5/25      |
>   | GPT-5.1            | 25/25  | 3/25      |
>
> That is, we found that *all* high-fidelity feature descriptions are accurate, while the *low-fidelity feature descriptions are much less reliable, which is why we excluded them from the analysis in our paper*. This experiment helps validate that our fidelity metric is a good judge of whether a feature description is accurate.
>
> Separately, in the paper, we also performed a human expert validation of the identified features that were high-fidelity and predictive of preferences. Out of 47 such features, 41 were rated helpful and all 47 were rated interpretable. Full details are in Appendix C.2.
>
> *The method drops prompts [...] but many preferences are conditional on the prompt. [...] Maybe add prompt-aware features, counterfactual tests, and prompt-conditioned regressions*.
>
> We agree with you that prompts could be important. To assess this, we ran your suggested experiment by encoding prompt information in two ways:
>
> 1. Instead of just embedding responses, embed the concatenation of the prompt P and response R. So, we predict preferences *y* using embed(P + R_A) - embed(P + R_B), where ‘+’ is string concatenation.
> 2. As another approach, we tried a regression that included prompt-response interaction terms. To do this, we predicted preferences *y* using the response features, but also every pairwise interaction between the response features and the prompt features. Since the embedding is 1536-dimensions, we first reduced the prompt and response embeddings to 32 dimensions with PCA and then included the 32*32 = 1024 interaction terms.
> The results are below:
>
> | Method | Avg | ChatbotArena | PKU | Reddit | HH-RLHF | CommunityAlign | PRISM | Tulu |
>   | --- | --- | --- | --- | --- | --- | --- | --- | --- |
>   | Response | 0.717 | 0.712 | 0.767 | 0.770 | 0.647 | 0.702 | 0.674 | 0.747 |
>   | Concat(Prompt, Response) | 0.722 | 0.715 | 0.776 | 0.765 | 0.646 | 0.715 | 0.690 | 0.744 |
>   | Response + P-R interactions  | 0.726 | 0.714 | 0.781 | 0.774 | 0.674 | 0.705 | 0.676 | 0.757 |
>
> As you can see, the preference predictions with the response-only achieve nearly the same AUC as the predictions with prompt-response interactions - the difference is less than 0.01 AUC, on average.
>
> This experiment suggests that intuitive ways of encoding the prompt don’t yield clear wins. However, it’s still possible that with a more sophisticated encoding of the prompt, we would see larger benefits, and so we will still further emphasize this opportunity on L141 and in Section 7’s future work paragraph.
>
> *Interpretable features recover only ~⅔ of a reward model’s AUC gain on average, and much less on HH-RLHF, suggesting important signals are still missing.*
>
> Yes, good point. For HH-RLHF specifically, it seems like prompt-response interactions might be more important, as including the interaction terms in the above experiment improves AUC from 0.647 to 0.674. On other datasets, WIMHF gets very close to the reward model (e.g., 0.700 AUC on ChatbotArena for SAE vs. 0.716 for the reward model). We hope that our paper spurs more work on methods that close the gap between interpretable features and the reward model, and we will describe this opportunity for future work in Section 7.

---

> ### Author Response · Authors · 2025-11-20
> **Author Response to AVFK (2/2)**
>
> *The label-flipping demo uses one small reward model and one main safety benchmark, then infers large leaderboard shifts. It would be better to validate across model scales/family and include one more safety benchmark.*
>
> Thanks for this suggestion - in light of your comment, we conducted the same experiment on (a) additional reward models of different size (Llama-3.2-1B) and model family (Qwen3-0.6B), and (b) on an additional dataset, RewardBench-1. RewardBench-1 has a different evaluation set than RewardBench-2, and is one of the only other high-quality reward modeling benchmarks.
>
> We found the same overall result with these new models and datasets: safety performance *substantially* increases (e.g., 4.2% to 46.2% for Llama-3.2-1B on RB-2) after flipping the top 1000 unsafe examples on ChatbotArena.
>
> | Model | RewardBench-1 |  | RewardBench-2 |  |
> |-------|---|---|---|---|
> | | No-Flip | Flip | No-Flip | Flip |
> | Qwen3-0.6B | 35.5% | 60.3% | 6.7% | 61.6% |
> | Llama-3.2-1B | 35.4% | 57.7% | 4.2% | 46.4% |
>
> *The pipeline removes non-English/very long items and even drops rows where both responses are marked “subjective” by an LLM[...]*
>
> To clarify, we *include* only the rows where either response is marked subjective by an LLM. This is because the goal of our paper is to study subjective human values, so we remove objective queries (like math and coding questions) to focus more on subjective preferences.
>
> We expect our method to generalize well to multilingual data, as long as the embedding models and LLMs perform well in these languages. For our study, we removed non-English data to enable more direct qualitative validation by the authors and the expert annotators. For example, it would be difficult for us to validate whether the LLM feature descriptions are high-quality in a language we cannot read. However, this is a limitation of the current work, as it’s possible that some preferences are language-specific, and we will mention this limitation in the Preprocessing section (L739).
>
> Long items (>2,048 tokens) were a tiny fraction of the data, less than 1%, and should not meaningfully affect the results; we simply removed them to avoid edge cases where text did not fit in the embedding model or LLM context windows.
>
> *Personalization is promising but thinly evaluated[...]*
>
> Yes, this is a fair point. Our main limitation in studying personalization was that only one dataset (Community Alignment) contained enough user-specific data to test for personalization, and even then, there is limited data per annotator. WIMHF should in principle work for multiple features, but the dataset as-is doesn’t have enough data spanning many different features, and we will flag this as a limitation in Section 7.
>
> *How robust are the learned features to the choice of LLM used for description and fidelity checks?*
>
> The fidelity scores tend to improve with the quality of the LLM used for description: for example, using gpt-5 instead of gpt-4.1 increases the average fidelity correlations from r=0.35 to r=0.40 on PRISM. We observe that the highest-fidelity features, like “mentions being an AI language model” (r=0.93) are very robust to different LLMs, while other features may vary more depending on the LLM. This is a promising finding, as it suggests that the results of the WIMHF method will improve as LLMs improve.
>
> *In the evaluation and Table 1, why don’t the colors (preferred vs. dispreferred) align with the “∆win” values? Do the colors indicate statistical significance?*
>
> Yes, the grey rows are features that do not reach significance after multiple testing correction. We included a legend on L221, and we will also clarify this point in the caption.
>
> ### Summary
>
> We believe the aforementioned experiments and clarifications have addressed your concerns. Given the strengths you highlight, and our experiments to address your comments, would you consider raising your score? If not, do you have further questions?

---

### Official Review · Reviewer_JHPE · 2025-10-29

**Soundness:** 3
**Presentation:** 2
**Contribution:** 3
**Rating:** 8
**Confidence:** 5

**Summary:**

How does preference optimization change language models, exactly? This paper trains SAEs contrastively between preferred and dispreferred pairs of responses to extract features underlying human preference. A language model is prompted with corresponding exemplars to generate human-readable summaries of these differences. The paper presents these summaries for several conversational datasets, also showing that human preference may differ or even oppose preferences collected from other sources or in other settings. Finally, the authors rewrite toxic examples identified from these preferences to show how they may be put to practical use.

**Strengths:**

This paper represents a rare case of an actual practical application of methods from Mechanistic Interpretability to the broader sphere of influence of LLMs. I can list several strengths here:

1. The overview of related work is both comprehensive and educational.
2. The figures look nice, and the paper is well written. I had no trouble following the writing, or understanding the examples.
3. The measurement of the variation in human preferences collected from different sources shows that people from different groups might prefer contradictory values—while not entirely surprising, this insight was unexpected for me.

I commend the authors on a well-written paper.

**Weaknesses:**

I only found two minor weaknesses in the paper.

1. At this moment, there is no intrinsic evaluation of how good the feature extraction pipeline is—specifically, you have shown good precision of extracted features, but not good recall. I don’t expect many changes for the review period, but I would bring up a potentially helpful experiment: why not create synthetic datasets where two responses differ precisely in some trait like “Likes to include anecdotes about pirates,” or other clearly identifiable traits, and see how many your pipeline can recover?
2. While the results on rewriting toxic examples are interesting, I have some concerns about whether such an approach can work more broadly (i.e., are there other examples of how identified traits can be used to modify or filter data?)

**Questions:**

I have one question for the authors. I had previously attempted to train SAEs contrastively to extract features at some point, but I recall that in my experiments, I found that the exemplars repeatedly led to the judge model (I believe it was GPT-4o at that point) outputting cookie-cutter explanations like “It is more helpful and comprehensive.” How did you manage to get precise explanations for the features? Did it require any prompt engineering? Or do you think using BatchTopK-SAEs allowed the extraction of more precise features where prompt engineering wasn’t required?

---

> ### Author Response · Authors · 2025-11-20
> **Author Response to JHPE**
>
> Thanks for the very kind and positive review! We were particularly delighted by your note that WIMHF is a “rare case of an actual practical application of methods from Mechanistic Interpretability to the broader sphere of influence of LLMs”. And we’re pleased that the paper was well-written and easy to follow.
>
> Thanks, also, for the helpful feedback: we ran an interesting semi-synthetic experiment as per your suggestion, described below. The experiment worked well, and we believe this helps address your concern about feature recall.
>
> *I would bring up a potentially helpful experiment: why not create synthetic datasets where two responses differ precisely in some trait[...]*
>
> We took 10,000 prompts (from the Community Alignment dataset), and for each prompt, we generated one default response with gpt-5-nano as well as one response with a persona as a system prompt, as you suggested:
>
> Personas:
> 1. **Pirate storyteller:** “Respond as a playful pirate who inserts short anecdotes about pirates in every answer.”
> 2. **Optimistic coach:** “Answer like an upbeat life coach who constantly cheers on the reader.”
> 3. **Nature poet:** “Describe everything using nature metaphors and lyrical language.”
> 4. **Sports commentator:** “Reply as if narrating a live game with play-by-play energy and sports metaphors.”
> 5. **Medieval scholar:** “Respond in archaic language as a learned monk who references ancient texts and philosophy.”
>
> We chose one of these personas randomly, and always labeled the persona response as the preferred response over the default response. Then, we ran WIMHF to output the top 5 features with largest coefficients. The results closely map onto the personas above:
>
> - Feature 3: "speaks in a pirate persona with nautical slang and metaphors"
> - Feature 6: "uses an upbeat, motivational coaching tone with affirmations"
> - Feature 1: "uses poetic, nature-inspired metaphors instead of straightforward, practical language"
> - Feature 5: "provides a numbered, step-by-step outline instead of a fully written draft"
> - Feature 7: "uses archaic, Shakespearean-style language"
>
> While real-world data is obviously more complex, we found this validation useful, and we hope it helps address your concern; thank you for suggesting this experiment!
>
> *I have some concerns about whether such an approach can work more broadly. (i.e., are there other examples of how identified traits can be used to modify or filter data?)*
>
>
> Yes. While the toxicity example was most clear, we do think that WIMHF identifies other actionable features. For example (L321), we observed that in Community Alignment (CA), responses focusing on sustainability and environmental concerns are almost always dispreferred. We likely wouldn’t want to encode this value in our models. Looking at the data, we hypothesize why this pattern emerges: in many examples, the model mentions sustainability even when it’s not relevant to the prompt.
>
>
> This sort of finding motivates an intervention where model developers could evaluate whether a model trained on CA learns to avoid discussing sustainability, and if so, they can explicitly curate data that helps address this. For example, they could generate data where the prompt *does* ask about a sustainable recommendation and the response provides it.
>
>
> Another example we discuss is the dispreference for clarifying questions. This may also be unideal: models *should* ask for clarification if necessary, though in practice they do not [1]. In response, dataset creators could explicitly collect examples with vague prompts and, correspondingly, generate responses with high-quality clarifying questions. This would teach the model to ask clarifying questions when appropriate, rather than always avoiding them.
>
>
> [1] Relying on the Unreliable: The Impact of Language Models' Reluctance to Express Uncertainty https://arxiv.org/abs/2401.06730
>
>
> *I recall that in my experiments, I found that the exemplars repeatedly led to the judge model (I believe it was GPT-4o at that point) outputting cookie-cutter explanations like “It is more helpful and comprehensive.”*
>
>
> Interesting! Did you use a pretrained SAE (like GemmaScope) or did you train your own?
> We found that training an SAE from scratch on the dataset(s) of interest was important to producing specific & interpretable latents. But other small tricks in training the SAE (like an auxiliary loss for dead features, BatchTopK, Matryoshka), as well as other differences in autointerp (prompt, exemplar sampling, etc.) may also explain the difference.
>
>
> When our paper is public, we would be happy for you to try out our code, and we can definitely chat further if it’s helpful.
>
>
> ### Summary
>
>
> Please let us know if you have any further questions! If we have addressed your concerns, we were wondering if you would consider raising your score? Thanks again for your questions and feedback.

---

> > ### Comment · Reviewer_JHPE · 2025-11-21
> > **Thank you for the rebuttal**
> >
> > Thank you for the response! The new experiment is quite convincing. I had originally trained my own SAEs on Llama-3.1-8B-Instruct, but I recall that I did not spend much time on the tricks mentioned in your response (I don't even think I had an auxiliary loss for dead features).
> >
> > I believe my concerns are now addressed. As I already gave the paper a high score, I will keep it. Once again, congratulations on a well-written paper.

---

> > > ### Author Response · Authors · 2025-11-21
> > > **Thank you!**
> > >
> > > Happy to hear you found the new experiment convincing, and makes sense regarding your prior results. Thanks again for the very kind review, cheers!

---

### Official Review · Reviewer_8ebG · 2025-10-30

**Soundness:** 3
**Presentation:** 4
**Contribution:** 4
**Rating:** 8
**Confidence:** 4

**Summary:**

This work focuses on the problem of interpreting human feedback, specifically finding explanations without pre-specifying hypotheses. Unlike previous approaches to this problem, this work uses sparse autoencoders (SAEs) on top of response embeddings to find features that human annotators may have followed. The authors refer to this method as What's In My Human Feedback (WIMHF). The authors demonstrate their method's effectiveness across a diverse set of datasets, including Chatbot Arena, Community Alignement, HH-RLHF, and more. They also illustrate how their method can be used for a number of downstream applications, such as improving safety of models, excluding misaligned preferences from evaluations, and personalisation.

**Strengths:**

This is a well written paper, that exhibits a number of notable strengths:

1. **Introduces a novel approach to feedback interpretation problem.** The problem of understanding human feedback remains very timely and important, given the importance of such datasets across training and evaluation pipelines. The approach introduced by the authors improves on prior work tackling the same problem (Inverse Constitutional AI, ICAI) in multiple ways:
    - Adds notion of measurable preferences in addition to realised preferences: Beyond just considering explaining the preference expressed by human annotators, this work also provides information about what can be preferred in a given dataset of prompt + response pairs. (Caveats apply as discussed in weaknesses)
    - Improves efficiency: using embedding models with a small SAE model is far less costly than the full-scale LLMs in prior work.
    - According to the experimental results, improves number of relevant features found over prior work, demonstrated with a number of interesting new insights.
2. **Convincingly demonstrates utility in extensive experiments.**  The results illustrate that their method is applicable across a wide range of datasets and use-cases, e.g. recovering known preferences and finding conflicting preferences across datasets. The experimental details are fully provided and the analysis thorough. The scope of these experimental results extends beyond prior work and thereby helps inform future work in this direction.
3. **Well written and executed.** Overall the paper is mostly clear and straightforward to follow, though some experimental discussions are very dense and rely heavily on the appendix. The experiments are well motivated and executed.

**Weaknesses:**

The work has a few weaknesses to consider:

1. **Correction for length seemingly arbitrary.** The approach currently corrects for length. There are a number of other well-known biases with human/ai annotators (e.g. position bias, preference for structured outputs, formatting, emojis etc.). How come you chose to correct for length only? This decision appears arbitrary and adds complexity to the method (Step 3 specifically). Shouldn't the length be automatically detected as one of the features? At least the decision to correct for length but not other features should be explicitly justified.
2. **Missing discussion of important limitations.** The proposed method (and prior approaches for the feedback data interpretation problem) have important limitations that are currently not explicitly discussed as far as I can see. In particular, the following limitations are not discussed but are important for downstream users:
	1. **Measurable (and realized) preferences are not exhaustive.**  There are multiple factors that may prevent the method from finding all theoretically avaialble measurable preferences: (1) the current experiments fix the number of features extracted by the SAE (in Step 1) to a small number, (2) the embedding models may not be able to capture very complex features of the responses, (3) the feature may not be extractable from embedding differences, and (4) the measurable preferences may depend on the prompt (which is not used in current experiments). These limits should at least be acknowledged.
	2. **Natural language preferences descriptions are not unique.**  The work fails to acknowledge that the natural language descriptions in Step 2 of features are not unique. An SAE feature may be explained by many different natural language descriptions. In particular, if two natural language features consistently correlate (e.g. using bold and italics formatting), the interpretation of an SAE feature is fundamentally ambiguous. Providing results re-running the pipeline and comparing the resulting features would be useful to estimate the impact of this and the previous limitation.
	3. **Correlation $\neq$ causation in realized preferences.**  Due to the previous point, as is common in interpretability work, the method can only show that human preferences correlate with the discovered realized preferences, rather than that human preferences *are* these realized preferences. Using the same hypothetical example as before, the annotators may only care about bold text but the method may declare that they care about italics text, and vice versa. This distinction is subtle but overconfidence can mislead users of the approach into incorrect beliefs about their annotators. The authors explicitly state that annotators "prefer" or "disprefer" (e.g. L295) certain things, which may be misleading without acknowledging this limitation. *Note that whilst this limitation is important to acknowledge, it does not strongly negatively impact the utility of the method (e.g. the improving the safety of trained models certainly still holds). Overall, the results should simply be viewed with awareness of the limitations.*
3. **No tie predictions.**  With the binary logistic regression as described in L165, tie votes are currently not considered. Nevertheless understanding the nature of ties, what leads to ties, also seems to be an interesting aspect of pairwise feedback -- that may be relevant to model developers trying to avoid or understand ties.
4. **Measurable preferences not directly comparable across datasets.** The measurable preferences rely on an SAE trained per-dataset: a feature is deemed measurable if the SAE learns it, otherwise it is not. This leads to dataset-specific features. Across datasets, these features may be similar but are not fully directly comparable/equivalent (not the same SAE). A direct comparison would make the results in Section 4.2 quite a bit stronger. This limited comparability with respect to measurable preferences appears to be a fundamental part of the approach in its current form.
5. **Limited comparison to prior work.** Currently the comparison to prior work originally introducing the feedback interpretation problem (ICAI) is limited to comparing the number of statistically significant features generated by each (and related analysis). Whilst this comparison is relevant, the authors omit a direct comparison of the overall combined reconstruction capabilities of the features generated by the two approaches, across different datasets. Such an analysis would be insightful and help make it easier to compare future approaches on this problem. Further, in the current comparison the prior work seems to be negatively affected by assumptions about the feature generation problem, assuming non-redundancy and no correlation with length, which are not necessarily objectives directly considered in the prior work. Thus, both uncorrected and length-corrected results would be interesting to understand the differences.
6. **Incorrect claim in L483:** This claim about no prior work being interpretable and data-driven does not appear to be true - ICAI seems to be a prior method that is both. (*"Unlike prior methods to understanding feedback data, WIMHF’s approach is both interpretable and data-driven, enabling discovery of new hypotheses without pre-specifying attributes to measure."*)

Overall, I consider this work to be of solid quality with high impact and, if the weaknesses are adequately addressed, may be suitable for spotlight or other recognition.

*Minor (no impact on score, no need to respond):*
1. In L164: would be good to have a number for this equation

**Questions:**

1. Generally, for each limitation/weakness above, I would be curious whether you agree and, if, how you are planning to address the concerns.
1. For the logistic regression in L164, the label $y$ is assumed to be binary, either response A or response B, correct? This could be clarified.
	- How would you adapt the proposed approach to allow for predicting tie votes (e.g. both good, both bad)?
2. How does the method ensure that features $z_i$ are not similar? In Appendix A.1 this is briefly discussed, but I would be curious how larger values of $M$ impact the results and lead to duplicates and redundancy?
3. How consistent are features across different training runs of the SAEs under the same settings?
4. Would you be able to extend on the rationale for excluding queries with objective correct answers? A lot of important observed biases are also applicable or only observable across such queries (e.g. preferring confident answers over truthful answers).
5. L301 onwards: Why don't you apply multiple SAEs across datasets? I don't understand why you can't transfer the SAEs directly: trained on dataset A and then used to understand the realised preferences on dataset B. Is there something fundamental preventing such a transfer?
6. Are you planning to release the code?

---

> ### Author Response · Authors · 2025-11-20
> **Author Response to 8ebG (1/4)**
>
> ### Overview
>
> Thank you for your kind, positive, and constructive review of our paper - we really appreciate it! We’re glad that you found our approach “novel”, “timely and important”, and useful “across a wide range of datasets and use-cases”.
>
> We very much appreciate your detailed questions and comments on possible limitations. We ran four experiments to help answer your questions concretely: (1) removing the length control and showing that the results change in an expected way, (2) showing that learned features are reproducible across runs, (3) comparing the dataset-specific SAE approach to the joint SAE approach, and (4) studying how the SAE’s M hyperparameter influences feature redundancy.
>
> We will describe these below inline, in addition to other clarifications, which we hope have collectively addressed your concerns!
>
> ### Responses
>
> *Correction for length seemingly arbitrary. [...] How come you chose to correct for length only? [...] Shouldn't the length be automatically detected as one of the features?*
>
> This is a good point with two sub-questions:
>
> **“Shouldn’t length be automatically detected”** -
> Yes: we conducted an experiment to explicitly compare the results of our method with and without length correction. Features that relate to length change in expected ways with and without the correction:
>
> Without length correction, on Reddit for example, the feature “provides a longer, multi-sentence explanation rather than a brief quip or single concise answer” is preferred. After we control for length, this feature is actually *dispreferred* - suggesting that, while Redditors prefer longer responses in general, for a given length, users prefer more quippy and concise language. This anecdote illustrates that the length preference emerges organically if we don’t control for it, but also that controlling for it can reveal findings that would’ve otherwise been “dominated” by length.
>
> On the other hand, on Community Alignment, length on its own does not correlate with preferences. To that point, we find that the feature win-rates are very similar with and without controlling for length - for example, a feature on using “narrative paragraphs instead of organized lists or bullet-pointed answers” has a win-rate delta of approximately -50% with or without controlling for length.
>
> **“How come you chose to correct for length only”**-
> (a) length is the most well-studied example of a feature that often predicts preference, making it a good candidate for a control. (b) we chose not to include other controls, like the preference for structured outputs, because we wanted to see if our method could automatically discover them, which it did.
>
> We will clarify in Step 3 of the method description (L167) that users need not control for any features or could control for aspects besides length, depending on their needs.
>
> *Measurable (and realized) preferences are not exhaustive.*
>
> This is an excellent point, and we will explicitly call it out in Section 7 (in which we’ll add a paragraph titled “Limitations and Future Work”). In particular, we will explain reasons why some features may be missed, as you point out: not using the prompt, the embeddings failing to capture some info, and the SAE losing info from the embeddings.
>
> In response to this concern, we ran an additional validation (which R3 also asked about, and it is described in full detail there). We used 5 system prompts as “ground-truth” preferences, and we found that WIMHF exactly recovered all 5 of these prompts (up to a paraphrase) as its top 5 features. Real-world preferences are more complex, but we think this experiment is still a helpful validation that our methodology can correctly recover preferences in some circumstances, in spite of the aforementioned limitations.

---

> ### Author Response · Authors · 2025-11-20
> **Author Response to 8ebG (2/4)**
>
> *Natural language preferences descriptions are not unique.*
>
> This is also a good point, and we will mention the limitation of imperfect/ambiguous feature descriptions as a new paragraph in Section 4, immediately prior to describing our results.
>
> That said, we have a few responses:
> 1. **Fidelity helps remove inaccurate descriptions:** Our main safeguard against inaccurate descriptions is our fidelity metric: we fully exclude all low-fidelity descriptions from our analysis. In response to Reviewer 1, we also did a human validation to further show that fidelity is very good at separating accurate descriptions from inaccurate ones.
> 2. **Ambiguous features:** By manually verifying several features, we found that if a single SAE feature seems to fire on multiple concepts, the LLM’s description usually mentions both - e.g. there is a feature on ChatbotArena that fires on headings, lists, and bold text, and our description mentions all of these.
> 3. **New experiment on feature consistency:** As per your suggestion, we directly evaluated the similarity of the feature descriptions generated by two trials of the method using different random seeds. We found that for a randomly-chosen feature in trial A, it generally has a nearest neighbor in trial B that is highly similar (we found these nearest neighbors by taking the max cosine similarity of the features’ text embeddings). For example, see below for 5 random features from HH-RLHF. All features from run A have a very similar analogue in run B.
>
> | run A feature | most similar run B feature |
> | --- | --- |
> | asks for more details or clarification instead of giving substantive advice | asks clarifying questions instead of directly providing substantive information |
> | questions the user's intentions and motives instead of addressing the request | does not directly address the request, instead asking questions or expressing confusion |
> | does not provide concrete, actionable guidance for committing wrongdoing | refuses to provide advice for committing violence or murder |
> | refuses to provide insulting or harmful content | refuses to provide harmful, offensive, or policy-violating content |
> | does not straightforwardly answer or provide concrete advice | withholds substantive answers (responds vaguely or with meta questions) |
>
> *Correlation $\ne$ causation in realized preferences.*
>
> Thank you for mentioning this; we agree we should more clearly differentiate between correlation and causation, and have made the following changes: (1) We are more careful about causal language throughout the paper, removing words like “affects” and replacing them with phrasing like “tends to increase predicted win-rate”. (2) We added the following limitation as a footnote in Step 3 of Section 3: *“Note that we cannot be sure if these features causally affect human preference. Rather, we are describing response features that correlate with annotator choices.”*
>
> *No tie predictions.*
>
> Yes, we will highlight ties in our new Future Work paragraph in Section 7. A main reason we didn’t study ties in this paper was that 6 out of our 7 datasets do not include them (only ChatbotArena had ties). However, we agree that it would be an interesting direction in future work to study what leads to ties.

---

> ### Author Response · Authors · 2025-11-20
> **Author Response to 8ebG (3/4)**
>
> *Measurable preferences not directly comparable across datasets. The measurable preferences rely on an SAE trained per-dataset[...]*
>
> This is a good question, and we’ll further explain our thinking here! During the development of WIMHF, using a single, joint SAE was our first approach - we shared your intuition that this would make dataset comparisons easier. However, we found it didn’t work as well. To respond to your comment, we ran another experiment to validate this earlier finding.
>
> The experiment revealed two main issues with the joint SAE:
>
> 1. **Feature splitting across datasets.** Even though datasets do have related features, in practice, the joint SAE learns separate features for separate datasets. For example, Arena, HH-RLHF, and PKU all have lots of “refusals” where the LLM declines to answer a harmful request. However, the joint SAE ends up learning separate refusal features per-dataset, because the exact “type” of refusal is heavily influenced by the exact prompts, which LLM generated the refusal, etc. So, there isn’t a single refusal feature that’s directly comparable across datasets. More nuanced features, like expressing uncertainty in the response, exhibit this splitting issue even more.
> 2. **Losing nuanced features.** The joint SAE loses some of the more nuanced dataset-specific features, which are some of the most interesting outputs of WIMHF. For example, PRISM has lots of prompts about the Israel-Palestine conflict, but other datasets do not. The joint SAE doesn’t learn an Israel-Palestine feature, while the PRISM-specific SAE does. One way this manifests is that, for most datasets, the dataset-specific SAE features yield better preference prediction AUC than the joint SAE. See below:
>
> | Model | Avg. | ChatbotArena | PKU | Reddit | HH-RLHF | CommunityAlign | PRISM | Tulu |
> | --- | --- | --- | --- | --- | --- | --- | --- | --- |
> | Joint SAE (M=100) | 0.671 | 0.681 | 0.711 | **0.720** | 0.604 | 0.655 | 0.664 | 0.659 |
> | Joint SAE (M=200) | 0.672 | 0.684 | 0.707 | 0.716 | **0.607** | 0.655 | 0.662 | 0.685 |
> | Dataset SAE | **0.684** | **0.700** | **0.732** | 0.717 | 0.587 | **0.687** | **0.671** | **0.696** |
>
> It’s possible that with more iteration, the joint SAE could work - we agree this is an interesting direction for future work - however, we did experiment with it fairly extensively, and found that the method in the paper yielded better results.
>
> *Limited comparison to prior work. Currently the comparison to prior work originally introducing the feedback interpretation problem (ICAI) is limited to comparing the number of statistically significant features generated by each (and related analysis).*
>
> Yes, fair point - in general, it is difficult to design good quantitative metrics to compare different methods, because the main objective of feedback interpretation is to surface practically actionable insights. In particular, AUC is not necessarily informative: a single feature like “the response is better” would achieve very high AUC if evaluated by an LLM judge, even though it would not be useful to researchers studying human feedback.
>
> This is why, besides the number of statistically significant features, we focused on qualitative differences: we found that WIMHF finds several important features that ICAI does not find, like the unsafe preference on ChatbotArena, or the dispreference for responses that mention the environment on Community Alignment. Ultimately, after analyzing the outputs of both methods across several datasets, we conclude that they can complement each other, because they surface different kinds of features, as discussed in L805-812. We will expand this discussion further in light of your comment.
>
> *Incorrect claim in L483[...]*
>
> Thanks for pointing this out - we’ve removed this claim.
>
> *For the logistic regression in L164, the label is assumed to be binary, either response A or response B, correct?*
>
> That’s right. We’ll clarify in L164 of the manuscript.
>
> *How would you adapt the proposed approach to allow for predicting tie votes (e.g. both good, both bad)?*
>
> In theory, WIMHF should be applicable to studying ties (e.g. by setting a label of 1 for any tie and 0 for any non-tie), though we haven’t explicitly tried this. Our method may require some changes to produce good results for ties.

---

> ### Author Response · Authors · 2025-11-20
> **Author Response to 8ebG (4/4)**
>
> *How does the method ensure that features are not similar? [...] I would be curious how larger values of M impact the results and lead to duplicates and redundancy?*
>
> Yes, the primary lever to control duplicates and redundancy is to edit the SAE hyperparameters. In light of your comment, we ran an experiment to test how M affects results, averaging across all 7 datasets and showing both (1) the % of non-redundant features and (2) the validation AUC using M = [16, 32, 64, 128].
>
> Link to the figure: https://imgur.com/a/GMhMJLA
>
> We found that increasing (1) M consistently reduces the percentage of features that are non-redundant and high-fidelity (from 39.3% with M=32 to 29.2% with M=128), and (2) AUC does not go up with a higher value of M.
>
> That being said, choosing M is still a judgment call, as M=128 leads to a much larger *total* number of non-redundant features vs. M=32 (e.g. 72/128 vs. 24/32 for Community Alignment). We found that M=32 strikes a good balance for the analysis in our paper. But ultimately it is up to the practitioner to choose how to weigh redundancy against possibly discovering a new feature of interest.
>
> We will add these quantitative results to our discussion in Appendix A.1.
>
> *How consistent are features across different training runs of the SAEs under the same settings?*
>
> Quite consistent! We replied to this above with some feature examples from HH-RLHF.
>
> *Would you be able to extend on the rationale for excluding queries with objective correct answers? A lot of important observed biases are also applicable or only observable across such queries (e.g. preferring confident answers over truthful answers).*
>
> This is a good point, and we will expand on our rationale on L180. The focus of our study is on subjective human preferences, so we filter for subjective conversations to increase the number of our findings that relate to these types of preferences.
>
> We do think our method could be applied to study all preference data, but it could require a slightly modified approach. One feature that would clearly matter is the correctness of the response, like whether a code snippet correctly implements a functionality. However, code correctness may not be captured by text embeddings - it may require more specialized code embeddings, along with some encoding of the prompt, or a different approach altogether (like using a judge LLM to evaluate whether the code is correct).
>
> For your example of studying correctness/truthfulness against other factors, we could therefore use a hybrid approach. We could use a judge LLM to annotate whether the response provides a correct answer to the prompt, and add this annotation as a covariate in Step 3. Then, we could use WIMHF to study what features matter to users on objective queries controlling for whether the response is correct (which might reveal that users care about confidence). This would be a cool direction for future work!
>
> *Are you planning to release the code?*
>
> Yes, thanks for asking - we have a ready-to-run code repository as well as our processed datasets posted to HuggingFace, and these will be linked in the public version of the paper. We are very much invested in making WIMHF a tool that is actually useful to practitioners!
>
> ### Summary
>
> Thank you again for your thorough review of our paper; we found it extremely constructive. The review has already been very beneficial for us to think through some of the more nuanced methodological points.
>
> We hope these experiments have addressed your concerns to your satisfaction, and we were wondering whether you’d consider raising your score? If not, is there anything that you would like to discuss further?

---

### Official Review · Reviewer_87Mp · 2025-11-01

**Soundness:** 2
**Presentation:** 2
**Contribution:** 3
**Rating:** 4
**Confidence:** 4

**Summary:**

This paper introduces What's In My Human Feedback (WIMHF), a novel method to automatically discover and interpret the preferences encoded in human feedback datasets. The technique operates by training a Sparse Autoencoder (SAE) on the differences between text embeddings of preference responses. This process identifies a small set of key features that differentiate the responses. These features are then translated into human-readable descriptions using an LLM. The authors demonstrate that WIMHF can successfully surface important, and sometimes harmful, preferences within data, such as a bias towards unsafe content. This allows for targeted data curation and a more transparent understanding of how preference data shapes model behavior.

**Strengths:**

+ The paper tackles the important and challenging task of understanding the implicit biases and preferences within human feedback data, distinguishing between what is "measurable" in the data and what is a "realized" human preference.

+ A key strength is the demonstrated ability of WIMHF to automatically detect misaligned or unsafe preferences in widely-used datasets, providing a practical tool for improving model safety.

**Weaknesses:**

- The paper does not provide sufficient ablation studies to justify each component of its methodology. It is unclear if every step, particularly the use of a Sparse Autoencoder (SAE), is necessary or if a simpler approach could achieve similar results.

- The experimental setup is often described with insufficient detail, making it difficult for readers to fully understand the settings and reproduce the results.

- Please see further questions.

**Questions:**

Could a simpler baseline, such as using an LLM to directly explain the preference between two responses and then clustering those explanations, achieve similar results? Is the SAE a necessary step?

How is the difference in text embeddings (e_delta) calculated? Is it the embedding of the chosen response minus the rejected one, or another way?

How do the results vary when using different text embedding models? Could the choice of the OpenAI text-embedding-3-small model introduce bias?

The features are described in natural language by an LLM. How can you be sure that the LLM's description is the most accurate or salient interpretation of that feature, and not just one of several possibilities?

The qualitative validation relies on experts rating LLM-generated feature descriptions. How does this validate the underlying features discovered by the SAE, rather than just the descriptive capability of the LLM?

For datasets generated by a limited number of LLMs (like Community Alignment), could the lack of response diversity artificially limit the types of features WIMHF can discover?

Could you clarify the rationale for filtering both subjective queries and objective ones (like math and coding) from the datasets?

Figure 4 indicates that the SAE method loses more information for datasets like HH-RLHF and Reddit. What properties of these datasets make their preferences harder to explain with interpretable features?

Could you provide more detail on how the "win-rate" metric is computed and on what specific datasets or splits it is evaluated?

---

> ### Author Response · Authors · 2025-11-20
> **Author Response to 87Mp (1/3)**
>
> ### Overview
>
> Thank you for your review! We’re glad you agree that the task of explaining human preferences is “important and challenging”, and that you appreciated our ChatbotArena application as “a practical tool for improving model safety.”
>
> Thanks also for posing several questions, which have helped us clarify and improve the paper. We have run three new experiments to address your concerns: (1) we ran two ablations of the SAE to demonstrate its value; (2) we showed that the learned features are robust to a different embedding model; (3) we conducted a human validation of our method’s feature descriptions.
>
> We believe that our new experiments and clarifications have addressed all your concerns.
>
> ### Responses
>
> *“The paper does not provide sufficient ablation studies to justify each component of its methodology. It is unclear if every step, particularly the use of a Sparse Autoencoder (SAE), is necessary or if a simpler approach could achieve similar results.”*
>
> Good point! To address your feedback, we ran an ablation removing the SAE. We tested 3 experimental conditions:
> 1. **SAE**: Our default method, which trains an SAE on embedding differences with M=32 features.
> 2. **Embed-TopDims**: Ablate the SAE, and interpret the embeddings directly. Since the embeddings have 1536 dimensions, we selected the 32 dimensions (to match SAE’s M) in the embedding differences that are most strongly correlated with the preference labels y. Then, we interpret these features in the embedding using the exact same procedure as for the SAE (Step 2 of our method on L148).
> 3. **Embed-PCA**: Similar to (2), but we use PCA to reduce the embeddings from 1536 dimensions down to 32, and then interpret these 32 features.
>
> Below, we show averaged metrics across all 7 datasets:
> | Method | Mean fidelity | High-fidelity features | Non-redundant high-fidelity features |
>   | --- | --- | --- | --- |
>   | SAE | **0.33** | **19.6 / 32** | **18.0 / 32** |
>   | Embed-TopDims | 0.20 | 12.7 / 32 | 10.7 / 32 |
>   | Embed-PCA | 0.13 | 4.9 / 32 | 4.6 / 32 |
>
> By all three metrics, the *SAE features are substantially more interpretable than the features produced by the ablations*: on average across the 7 datasets, our pipeline produces high-fidelity interpretations for 19.6 out of 32 features when using the SAE, as compared to 12.7 and 4.9 with the top embedding dimensions and embedding PCA, respectively.
>
> This demonstrates that the SAE is critical to produce interpretable features, which are the basis of the WIMHF pipeline. This finding is consistent with several prior works in interpretability that establish a similar result (e.g. [1] [2]), but we agree that running the ablation for our specific datasets is important, so thank you for this suggestion. We will add this table to the paper in Appendix A.3.
>
> [1] Disentangling Dense Embeddings with Sparse Autoencoders https://arxiv.org/abs/2408.00657
> [2] Sparse Autoencoders for Hypothesis Generation https://arxiv.org/abs/2502.04382
>
> *“Could a simpler baseline, such as using an LLM to directly explain the preference between two responses and then clustering those explanations, achieve similar results?”*
>
> Thanks for asking - using an LLM to explain the difference, and then clustering, is precisely how the Inverse Constitutional AI (ICAI) baseline works. We describe our comparison to this baseline on L457-460. We found that WIMHF finds more statistically significant preferences than ICAI, and, qualitatively, WIMHF finds several important features that ICAI does not find, like the unsafe preference on ChatbotArena, or the dispreference for responses that mention the environment on Community Alignment.
>
> *“How is the difference in text embeddings (e_delta) calculated? Is it the embedding of the chosen response minus the rejected one, or another way?”*
>
> $e_\Delta = \text{embedding}(r_A) - \text{embedding}(r_B)$. Either response A or response B may be chosen or rejected; we use a convention where if response A is chosen over response B, then the label y is 1; if response B is chosen over A, then the label y = 0.

---

> ### Author Response · Authors · 2025-11-20
> **Author Response to 87Mp (2/3)**
>
> *“How do the results vary when using different text embedding models? Could the choice of the OpenAI text-embedding-3-small model introduce bias?”*
>
> We agree it’s important that WIMHF is robust to the choice of the embedding model. To check this, we ran an experiment where we re-ran our pipeline using an open-source ModernBERT embedding model (nomic-ai/modernbert-embedding-base).
>
> We found that *using a different embedding model produces qualitatively very similar features*. In the below table, we are showing 5 randomly chosen features (not cherry picked) on HH-RLHF. For each feature randomly picked from the list of features produced using the OpenAI embeddings, we find its semantically closest feature from ModernBERT (by computing the cosine similarity of the feature description’s text embedding). Qualitatively, these feature pairs are very similar.
>
> | text-embedding-3-small | nomic-ai/modernbert-embed-base |
> |---|---|
> | avoids giving a substantive answer, expressing confusion or reluctance instead | deflects the user’s question by expressing confusion or lack of knowledge instead of providing a substantive answer |
> | does not provide concrete instructions, strategies, or example phrases for carrying out the requested harmful or rude action | responds briefly without providing specific, detailed information or concrete suggestions |
> | provides a concrete answer or advice in response to the user's request, rather than asking follow-up or clarifying questions | directly answers the user's request with substantive content instead of expressing confusion or asking for clarification |
> | explicitly references physical harm, punishment, or death | explicitly talks about harming, hurting, or causing suffering to others |
> | refuses to provide advice that enables harmful, illegal, or unethical behavior | refuses or discourages harmful, illegal, or unethical actions or beliefs |
>
> This result suggests that WIMHF would work well with any strong embedding model, consistent with prior work demonstrating that different embedding models learn similar semantic features (e.g. [1] [2]). We will add this result to Appendix A.
>
> [1] Harnessing the Universal Geometry of Embeddings https://arxiv.org/abs/2505.12540
> [2] Word Translation Without Parallel Data https://arxiv.org/abs/1710.04087
>
> *“The features are described in natural language by an LLM. How can you be sure that the LLM's description is the most accurate or salient interpretation of that feature, and not just one of several possibilities?”*
>
> Yes, good point. Our main measure of the “accuracy” of the LLM’s interpretation is the fidelity score, where we check the quality of the feature description using a separate annotator LLM. This approach is used in prior work [1] [2]. If the fidelity score is not statistically significant, then we *ignore the feature entirely*, mitigating the risk of inaccurate feature descriptions entering our analysis.
>
> To further address your concern, we conducted a human validation to check whether fidelity indeed helps distinguish accurate from inaccurate descriptions:
> 1. Take the 25 feature descriptions with highest fidelity + 25 with lowest fidelity, out of the 224 features total (7 datasets * 32 features/dataset).
> 2. Manually look at 5 sampled (prompt, response A, response B) triplets that were used to generate the feature description.
> 3. For the 50 total features (25 high-fidelity + 25 low-fidelity), randomly shuffle them, and then check whether the feature description accurately summarizes the data in (2).
>
> We had both a human and GPT-5.1 perform this analysis, with similar results. See results below:
>
> | Judge           | Top-25 Fidelity | Bottom-25 Fidelity |
>   | ---                | ---    | ---       |
>   | Human              | 25/25   | 5/25      |
>   | GPT-5.1            | 25/25  | 3/25      |
>
> That is, we found that *all* high-fidelity feature descriptions are an accurate summary of what the top-activating SAE exemplars share. The *low-fidelity feature descriptions are much less reliable, which is why we excluded them from the analysis in our paper*.
>
> Overall, this experiment helps validate that our fidelity metric is a good judge of whether a feature description is accurate. To make it easy for practitioners to filter on fidelity, we report all fidelity scores alongside the features in our full data/demo release. However, it is still a good point that the interpretations may not be perfect, and we added a footnote in the Methods section to clarify this.
>
> [1] Dami Choi et al. Scaling automatic neuron description, October 2024. https://transluce.org/neuron-descriptions.
> [2] Steven Bills et al. Language models can explain neurons in language models. https://openaipublic.blob.core.windows.net/neuron-explainer/paper/index.html, 2023.

---

> ### Author Response · Authors · 2025-11-20
> **Author Response to 87Mp (3/3)**
>
> *“The qualitative validation relies on experts rating LLM-generated feature descriptions. How does this validate the underlying features discovered by the SAE, rather than just the descriptive capability of the LLM?”*
>
> The SAE is valuable because it allows features to be described with high-fidelity. No matter how good the LLM, it wouldn’t be able to provide a good description of an inherently uninterpretable feature, as shown by our ablation experiments. The expert validation goes one step further and aims to demonstrate that not only do our features pass the filters of being high-fidelity and predictive, they are also helpful and interesting to experts.
>
> *“For datasets generated by a limited number of LLMs (like Community Alignment), could the lack of response diversity artificially limit the types of features WIMHF can discover?”*
>
> Interesting question! We observe that even though all responses on Community Alignment are generated by Llama-3.3-70B, we observe a substantial amount of feature diversity (Table 10). Qualitatively, it looks like Community Alignment has similar or more diversity along value and topic axes than other datasets (e.g. ChatbotArena), which contain many different LLMs.
>
> What’s useful about WIMHF is that it gives us a concrete procedure to assess a dataset’s response diversity, and how the choice of LLMs, sampling strategy, etc affect diversity (L274). These are questions that are difficult to answer without WIMHF.
>
> *“Could you clarify the rationale for filtering both subjective queries and objective ones (like math and coding) from the datasets?”*
>
> A slight correction: we didn’t filter out subjective queries; these were the examples that we *kept*. We filtered out prompts deemed to have *objective* answers, like “How to run python code in one line using zsh?”. The rationale behind this was that objective queries tend to have a clear reason for being preferred or not: whether the response is correct or incorrect. We were more interested in what humans prefer when “correctness” is not as objectively defined, which was why we performed this filtering.
>
> *“Figure 4 indicates that the SAE method loses more information for datasets like HH-RLHF and Reddit. What properties of these datasets make their preferences harder to explain with interpretable features?”*
>
> Great question - we’re not sure, but this would be an interesting avenue for future work! One possibility is that preferences in HH-RLHF depend more on the prompt, while our SAE is trained only on responses. Some evidence for this possibility is that fitting a regression with prompt-response interactions slightly helps AUC on HH-RLHF, unlike on other datasets. If this is true, it suggests that there is an opportunity for future improvements to the method via better handling of the prompt.
>
> *“Could you provide more detail on how the "win-rate" metric is computed and on what specific datasets or splits it is evaluated?”*
>
> To compute win-rate (see L171-173 and L729-734), we fit a regression $\mathbb{P}(y = 1) = \sigma(\alpha + \beta_j \cdot D(z_j) + \gamma \cdot \mathbf{x}),$ where $D(z_j) = +1$ if $z_j > 0$ and $0$ if $z_j < 0$; values with $z_j = 0$ are excluded.
>
> This enables computing the *average marginal effect*, $\sigma(\beta_j + \alpha + \gamma \cdot \mathbf{x}) - \sigma(\alpha + \gamma \cdot \mathbf{x})$, i.e., the mean change in win rate for positive vs. negative $z_j$ while holding length constant [1].
>
> [1] R. Williams. Using the Margins Command to Estimate and Interpret Adjusted Predictions and
> Marginal Effects. The Stata Journal, June 2012.
>
> Win-rate is computed for each feature on all datasets, with results in Tables 9-15.
>
> ### Summary
>
> Thanks again for all of your excellent suggestions and questions, which were very helpful in motivating new experiments and clarifications to the paper.
>
> Given the strengths you highlight, and our three new experiments which we hope have comprehensively addressed your questions, would you consider raising your score? If not, please let us know if you have any other questions, or if there are other experiments that might be helpful - thanks so much!

---

### Author Response · Authors · 2025-11-20
**Overall response & summary of changes**

Thanks to all reviewers for their thoughtful reviews, and we're glad that the overall impression of the paper is very positive!

In particular, **all reviewers** noted how the paper demonstrates practical applications for post-training, one reviewer (**JHPE**) noted how WIMHF is a rare case of a useful application of interpretability in language modeling research, and multiple reviewers (**8ebG**, **JHPE**) found the paper well-written and interesting.

We're also grateful for the suggestions and questions posed by the reviewers, which we've addressed carefully through several new experiments and clarifications.

We summarize some of the main new experiments below:

1. **Ablations:** In response to **87Mp**, we performed ablations where we remove the SAE and keep the rest of our pipeline constant. We found that without the SAE, the embedding features are much less interpretable, supporting the need for the SAE.
2. **Robustness evaluations:** In response to **87Mp** and **8ebG** respectively, we evaluated the robustness of our learned features to (1) using a different embedding model and (2) randomness due to data splitting and LLM stochasticity. We find that the features are highly robust to these changes.
3. **Additional human validation:** **87Mp** and **AVFK** asked for more human validation of our feature description pipeline, beyond the qualitative validation study already in the paper. We conducted a new validation of 50 generated features, 25 with high-fidelity and 25 with low-fidelity, and we found that fidelity was indeed very effective at separating good feature descriptions from bad ones. As a reminder, we exclude low-fidelity features from all analyses in Sections 4+5.
4. **Recovering ground-truth semi-synthetic preferences:** **JHPE** asked whether our method recovers preferences in a setting where we know exactly what humans prefer. We tested this with a new semi-synthetic experiment, and we found that WIMHF perfectly recovered all 5 ground-truth preferences.

We also ran a few additional experiments (such as the impact of the SAE's hyperparameters in response to **8ebG**), described in further detail in the respective replies.

Finally, in our replies, we've also described how we will change the language of the paper to address the reviewer's comments. Altogether, we believe that our new experiments and responses comprehensively address reviewer concerns.

---

### Meta-Review · Area_Chair_FSGG · 2026-01-06

**Summary:**

This paper introduces WIMHF, a method that uses sparse autoencoders on text embedding differences to discover interpretable features from human preference data. The authors demonstrate practical applications: flipping labels on unsafe examples yields substantial safety gains without degrading general performance, and identified features enable interpretable personalization. This is a rare practical application of interpretability methods to alignment. The main concerns centered on whether the SAE is necessary, reliability of LLM-generated feature descriptions, robustness to embedding model choice, and whether the method can recover ground-truth preferences. The authors ran extensive new experiments addressing each concern. Remaining limitations include the gap between interpretable features and full reward model performance (especially on HH-RLHF), limited prompt conditioning, and thin personalization evaluation due to data constraints.

**Reviewer Concerns:**

Reviewer 87Mp's concerns about SAE necessity were addressed through ablations showing SAE features achieve high-fidelity interpretations versus embedding baselines; concerns about feature description accuracy were addressed via human validation; embedding model robustness was demonstrated with ModernBERT producing qualitatively similar features.

Reviewer 8ebG's concerns about length correction arbitrariness were addressed by showing length preferences emerge organically without correction; limitations regarding non-exhaustive features and correlation-vs-causation were acknowledged with promised additions to Section 7; the joint-vs-separate SAE question was addressed experimentally, showing dataset-specific SAEs yield better AUC.

Reviewer JHPE's concern about feature recall was addressed through the semi-synthetic persona experiment recovering all ground-truth preferences.

Reviewer AVFK's concerns about LLM circularity were addressed via human audits; prompt-aware features were tested but showed minimal AUC improvement; reward model evaluation was extended to two additional models and RewardBench-1.

**Reviewer Scores:**

All reviewers would likely maintain their current scores.

---

### Decision · Program_Chairs · 2026-01-26

Accept (Oral)